# Improved Bayes Risk Can Yield Reduced Social Welfare Under Competition

**Meena Jagadeesan**
UC Berkeley
mjagadeesan@berkeley.edu

**Michael I. Jordan**
UC Berkeley
jordan@cs.berkeley.edu

**Jacob Steinhardt**[*]
UC Berkeley
jsteinhardt@berkeley.edu

**Nika Haghtalab**[*]
UC Berkeley
nika@berkeley.edu

## Abstract

As the scale of machine learning models increases, trends such as scaling laws anticipate consistent downstream improvements in predictive accuracy. However, these trends take the perspective of a single model-provider in isolation, while in reality providers often compete with each other for users. In this work, we demonstrate that competition can fundamentally alter the behavior of these scaling trends, even causing overall predictive accuracy across users to be non-monotonic or decreasing with scale. We define a model of competition for classification tasks, and use data representations as a lens for studying the impact of increases in scale. We find many settings where improving data representation quality (as measured by Bayes risk) decreases the overall predictive accuracy across users (i.e., social welfare) for a marketplace of competing model-providers. Our examples range from closed-form formulas in simple settings to simulations with pretrained representations on CIFAR-10. At a conceptual level, our work suggests that favorable scaling trends for individual model-providers need not translate to downstream improvements in social welfare in marketplaces with multiple model providers.

## 1   Introduction

Scaling trends in machine learning suggest that increasing the scale of a system consistently improves predictive accuracy. For example, scaling laws illustrate that increasing the number of model parameters [Kaplan et al., 2020, Sharma and Kaplan, 2020, Bahri et al., 2021] and amount of data [Hoffmann et al., 2022] can reliably improve model performance, leading to better representations and thus better predictions for downstream tasks [Hernandez et al., 2021].

However, these scaling laws typically take the perspective of a single model-provider in isolation, when in reality, model-providers often compete with each other for users. For example, in digital marketplaces, multiple online platforms may provide similar services (e.g., Google search vs. Bing, Spotify vs. Pandora, Apple Maps vs. Google) and thus compete for users on the basis of prediction quality. A distinguishing feature of competing platforms is that users can switch between platforms and select a platform that offers them the highest predictive accuracy for their specific requests. This breaks the direct connection between predictive accuracy of a single platform in isolation and social welfare across competing platforms, and raises the question: *what happens to scaling laws when model-providers compete with each other?*

---

[*]Equal contribution

37th Conference on Neural Information Processing Systems (NeurIPS 2023).

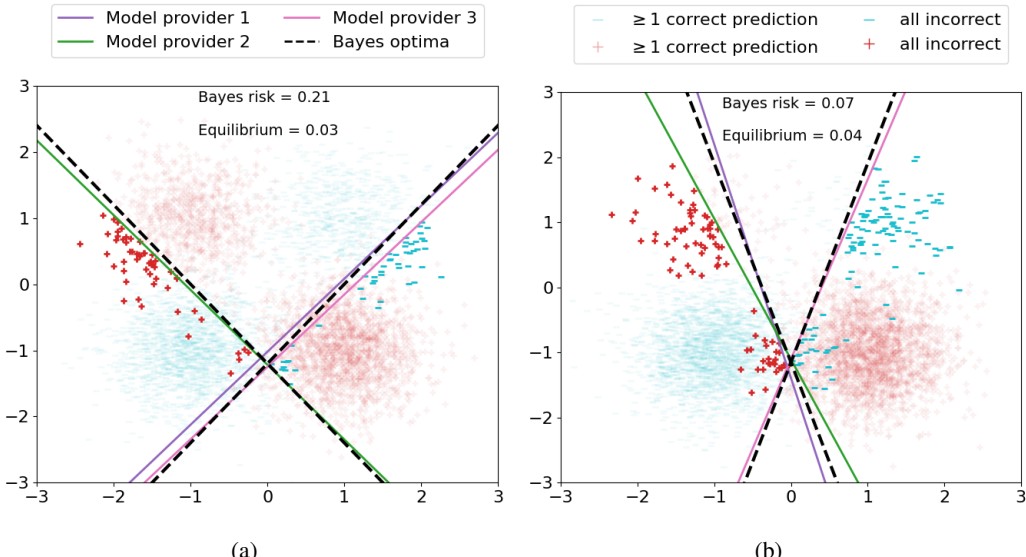

Figure 1: Comparison of equilibrium loss on two data distributions, one with high Bayes risk (left) and one with lower Bayes risk (right). Each plot shows the linear predictors chosen at equilibrium under competition between three model-providers (solid lines), along with two approximately Bayes-optimal predictors (dashed lines). The equilibrium social loss is lower in the left plot than the right plot, even though the Bayes risk is much higher. The intuition is that approximate Bayes optima disagree on more data points in the left plot than in the right plot; thus, users have a greater likelihood of at least one predictor offering them a correct prediction, which increases the overall predictive accuracy for users (i.e., the social welfare).

We show that the typical intuition about scaling laws can fundamentally break down under competition. Surprisingly, even monotonicity can be violated: increasing scale can *decrease* the overall predictive accuracy (social welfare) for users. More specifically, we study increases to scale through the lens of data representations (i.e., learned features), motivated by how increasing scale generally improves representation quality [Bengio et al., 2013].[2] We exhibit several multi-class classification tasks where better data representations (as measured by Bayes risk) *decrease* the overall predictive accuracy (social welfare) for users, when varying data representations along several different axes.

The basic intuition for this non-monotonicity is illustrated in Figure 1. When data representations are low quality, any predictor will be incorrect on a large fraction of users, and near-optimal predictors may disagree on large subpopulations of users. Model providers are thus incentivized to choose complementary predictors that cater to different subpopulations (market segments), thus improving the overall predictive accuracy for users. In contrast, when representations are high quality, each optimal predictor is incorrect on only a small fraction of users, and near-optimal predictors likely agree with each other on most data points. As a result, model-providers are incentivized to select similar predictors, which decreases the overall predictive accuracy for users.

To study when non-monotonicity can occur, we first focus on a stylized setup that permits closed-form calculations of the social welfare at equilibrium (Section 3). Using this characterization, in three concrete binary classification setups, we show that the equilibrium social welfare can be non-monotonic in Bayes risk. In particular, we vary representations along three axes—the per-representation Bayes risks, the noise level of representations, and the dimension of the data representations—and exhibit non-monotonicity in each case (Figure 2).

Going beyond the stylized setup of Section 3, in Section 4 we consider linear function classes and demonstrate empirically that the social welfare can be non-monotonic in the data representation

---

[2]We are motivated by emerging marketplaces where different model-providers utilize the same pretrained model, but *finetune* the model in different ways. To simplify this complex training process, we conceptualize pretraining as *learning data representations (e.g., features)* and fine-tuning as *learning a predictor from these representations*. In this formalization, increasing the scale of the pretrained model leads to improvements in data representations accessible to the model-providers during "fine-tuning".

quality. We consider binary and 10-class image classification tasks on CIFAR-10 where data representations are obtained from the last-layer representations of AlexNet, VGG16, ResNet18, ResNet34, and ResNet50, pretrained on ImageNet. Better representations (as measured by Bayes risk) can again perform worse under competition (Figure 3). We also consider synthetic data where we can vary representation quality more systematically, again finding ubiquitous non-monotonicities.

Altogether, our results demonstrate that the classical setting of a single model-provider can be a poor proxy for understanding multiple competing model-providers. This suggest that caution is needed when inferring that increased social welfare necessarily follows from the continuing trend towards improvements in predictive accuracy in machine learning models. Machine learning researchers and regulators should evaluate methods in environments with competing model-providers in order to reasonably assess the implications of raw performance improvements for social welfare.

## 1.1  Related work

Our work connects to research threads on the *welfare implications of algorithmic decisions* and *competition between data-driven platforms*.

**Welfare implications of algorithmic decisions.** Recent work investigates *algorithmic monoculture* [Kleinberg and Raghavan, 2021, Bommasani et al., 2022], a setting in which multiple model-providers use the same predictor. In these works, monoculture is intrinsic to the decision-making pipeline: model-providers are given access to a shared algorithmic ranking [Kleinberg and Raghavan, 2021] or shared components in the training pipeline [Bommasani et al., 2022]. In contrast, in our work, monoculture may arise endogenously from competition, as a result of scaling trends. Model-providers are always given access to the same function classes and data, but whether or not monoculture arises depends on the quality of data representations and its impact on the incentives of model-providers. Our work thus offers a new perspective on algorithmic monoculture, suggesting that it may arise naturally in competitive settings as a side effect of improvements in data representation quality.

More broadly, researchers have identified several sources of mismatch between predictive accuracy and downstream welfare metrics. This includes *narrowing* of a classifier under repeated interactions with users [Hashimoto et al., 2018], *preference shaping* of users induced by a recommendation algorithm [Carroll et al., 2022, Dean and Morgenstern, 2022, Curmei et al., 2022], *strategic adaptation* by users under a classifier [Brückner et al., 2012, Hardt et al., 2016], and the *long-term impact of algorithmic decisions* [Liu et al., 2018, 2020].

**Competition between data-driven platforms.** Our work is also related to the literature on competing predictors. The model in our paper shares similarities with Ben-Porat and Tennenholtz [2017, 2019], who studied equilibria between competing predictors. Ben-Porat and Tennenholtz [2017, 2019] show that empirical risk minimization is not an optimal strategy for a model-provider under competition and design algorithms that compute the best-responses; in contrast, our focus is on the equilibrium social welfare and how it changes with data representation quality. The specifics of our model also slightly differ from the specifics of Ben-Porat and Tennenholtz [2017, 2019]. In their model, each user has an accuracy target that they wish to achieve and randomly chooses between model-providers meeting that target; in contrast, in our model, each user noisily chooses the model-provider that minimizes their loss and model-providers can have asymmetric market reputations.

Our work also relates to *bias-variance games* [Feng et al., 2019] between competing model-providers. However, Feng et al. [2019] focus on the the equilibrium strategies for the model-provider, but do not consider equilibrium social welfare for users; in contrast, our work focuses on the equilibrium social welfare. The model of Feng et al. [2019] also differs from the model in our work. In Feng et al. [2019], a model-provider action is modeled as choosing an error *distribution* for each user, and the action set includes error distributions with a range of different variances. In contrast, in our setup, the error distribution for every user is always a point mass (variance 0). Thus, the equilibrium characterization of Feng et al. [2019] does not translate to our model. The specifics of the model-provider utility in the work of Feng et al. [2019] differs slightly from our model as well.

Other aspects studied in this research thread include competition between model-providers using *out-of-box* learning algorithms that do not directly optimize for market share [Ginart et al., 2021, Kwon et al., 2022, Dean et al., 2022], competition between model-providers selecting *regularization parameters* that tune model complexity [Iyer and Ke, 2022], competition between *bandit algorithms* where data directly comes from users [Aridor et al., 2020, Jagadeesan et al., 2022], and competition

between *algorithms dueling* for a user [Immorlica et al., 2011]. Our work also relates to *classical economic models of product differentiation* such as Hotelling's model [Hotelling, 1981, d'Aspremont et al., 1979] (see Anderson et al. [1992] for a textbook treatment), as well as the emerging area of *platform competition* [see, e.g., Jullien and Sand-Zantman, 2021, Calvano and Polo, 2021].

## 2 Model

We focus on a multi-class classification setup with input space $X \subseteq \mathbb{R}^d$ and output space $Y = \{0, 1, 2, \ldots, K - 1\}$. Each user has an input $x$ and a corresponding true output $y$, drawn from a distribution $\mathcal{D}$ over $X \times Y$. Model providers choose predictors $f$ from some model family $\mathcal{F} \subseteq (\Delta(Y))^X$ where $\Delta(Y)$ is the set of distributions over $Y$. A user's loss given predictor $f$ is $\ell(f(x), y) = \mathbb{P}[y \neq f(x)]$. In Section 3, we take $\mathcal{F} = \{0, 1, 2, \ldots, K - 1\}^X$ to be all deterministic functions mapping inputs to classes, while in Section 4 we consider linear predictors of the form $f(x) = \text{softmax}(Wx + b)$.

We study competition between $m \geq 2$ model-providers for users, building on the model of Ben-Porat and Tennenholtz [2017, 2019]. We index the model-providers by $[m] := \{1, 2, \ldots, m\}$, and let $f_j$ denote the predictor chosen by model provider $j$. After the model-providers choose predictors $f_1, \ldots, f_m$, each user then chooses one of the $m$ model-providers to link to, based on prediction accuracy. Model-providers aim to optimize the number of users that they win. We describe the model in detail below. (We note that this model is stylized and will make several simplifying assumptions; we defer a detailed discussion of the implications of these assumptions to Appendix A.)

**User decisions.** Users noisily pick the model-provider offering the best predictions for them. That is, a user with representation $x$ and true label $y$ chooses a model-provider $j^*(x, y)$ such that the loss $\ell(f_{j^*(x,y)}(x), y)$ is the smallest across all model-providers $j \in [m]$, subject to noise in user decisions. More formally, we model user noise with the logit model [Train, 2002], also known as the Boltzmann rationality model:

$$\mathbb{P}[j^*(x, y) = j] = \frac{e^{-\ell(f_j(x), y)/c}}{\sum_{j'=1}^{m} e^{-\ell(f_{j'}(x), y)/c}}, \tag{1}$$

where $c > 0$ denotes a noise parameter. We extend this model to account for uneven market reputations across decisions in Section B.1.

**Model provider incentives.** A model-provider's utility is captured by the market share that they win. That is, model-provider $j$'s utility is

$$u(f_j; \mathbf{f}_{-j}) := \mathop{\mathbb{E}}_{(x,y) \sim \mathcal{D}} [\mathbb{P}[j^*(x, y) = j]],$$

where $\mathbf{f}_{-j}$ denotes the predictors chosen by the other model-providers and where the expectation is over $(x, y)$ drawn from $\mathcal{D}$. Since the market shares always sum to one, this is a constant-sum game.

Each model-provider chooses a *best response* to the predictors of other model-providers. That is, model-provider $j$ chooses a predictor $f_j$ such that

$$f_j \in \arg\max_{f \in \mathcal{F}} u(f_j; \mathbf{f}_{-j}).$$

The best-response captures that model-providers optimize for market share. In practice, model-providers may do so via A/B testing to steer towards predictors that maximize profit, or by actively collecting data on market segments where competitors are performing poorly.

We study market outcomes $\mathbf{f} = (f_1, f_2, \ldots, f_m)$ that form a Nash equilibrium. Recall that $\mathbf{f}$ is a *pure strategy Nash equilibrium* if for every $j \in [m]$, model-provider $j$'s predictor is a best-response to $\mathbf{f}_{-j}$: that is, $f_j \in \arg\max_{f \in \mathcal{F}} u(f_j; \mathbf{f}_{-j})$. In well-behaved instances, pure-strategy equilibria exist (see Proposition 1 and simulation results in Section 4). However, for uneven market reputations (Appendix B.1), we must turn to mixed strategy equilibria where model-providers choose distributions $\mu$ over $\mathcal{F}$.

**Quality of market outcome for users.** We are interested in studying the quality of a market outcome $\mathbf{f} = (f_1, f_2, \ldots, f_m)$ in terms of user utility. The quality of $\mathbf{f}$ is determined by the overall *social loss* that it induces on the user population, after users choose between model-providers:

$$\text{SL}(f_1, \ldots f_m) := \mathbb{E}[\ell(f_{j^*(x,y)}(x), y)]. \tag{2}$$

When $f_1, \ldots, f_m$ is a Nash equilibrium, we refer to $\text{SL}(f_1, \ldots f_m)$ as the *equilibrium social loss.*

Our goal is to study how the equilibrium social loss changes when the representation quality (i.e., the quality of the input representations $X$) improves. We formalize representation quality as the minimum risk $\text{OPT}_{\text{single}}$ that a single model-provider could have achieved on the distribution $\mathcal{D}$ with the model family $\mathcal{F}$. This means that $\text{OPT}_{\text{single}}$ is equal to the Bayes risk:

$$\text{OPT}_{\text{single}} := \min_{f \in \mathcal{F}} \mathbb{E}\left[\ell(f(x), y)\right].$$

We show that the equilibrium social loss $\text{SL}(f_1^*, \ldots f_m^*)$ can be non-monotonic in the representation quality (as measured by $\text{OPT}_{\text{single}}$), when representations are varied along a variety of axes.

## 3 Non-monotonicity of Equilibrium Social Loss in a Stylized Setup

To understand when non-monotonicity can occur, we first consider a stylized setup (described below) that permits closed-form calculations of the social loss. We characterize the equilibrium social loss in this setup for binary classification (Proposition 2), and apply this characterization to three concrete setups that vary representation quality along different axes (Section 3.2): we show that the equilibrium social loss can be non-monotonic in Bayes risk in all of these setups (Figures 2b-2c). Finally, we extend our theoretical characterization in Proposition 2 to setups with more than 2 classes (Section 3.3), and we extend to model-providers with unequal market reputations (Appendix B.1).

**Specification of stylized setup.** Assume the input space $X$ is finite and let $\mathcal{F} = \mathcal{F}_{\text{all}}^{\text{multi-class}}$ contain all deterministic functions from $X$ to $\{0, 1, \ldots, K-1\}$. For simplicity, we assume that users make noiseless decisions (i.e., $c \to 0$), so a user's choice of model-provider $j^*(x, y)$ is specified as follows:

$$\mathbb{P}[j^*(x, y) = j] = \begin{cases} 0 & \text{if } j \notin \arg\min_{j' \in [m]} \mathbb{1}[y \neq f_{j'}(x)] \\ \frac{1}{\left|\arg\min_{j' \in [m]} \mathbb{1}[y \neq f_{j'}(x)]\right|} & \text{if } j \in \arg\min_{j' \in [m]} \mathbb{1}[y \neq f_{j'}(x)]. \end{cases} \tag{3}$$

In other words, users pick the model-provider with minimum loss, choosing randomly in case of ties. We show that pure strategy equilibria are guaranteed to exist in this setup.

**Proposition 1.** *Let $X$ be a finite set of representations, let there be $K \geq 2$ classes, let $\mathcal{F} = \mathcal{F}_{\text{all}}^{\text{multi-class}}$, and let $\mathcal{D}$ be the distribution over $(X, Y)$. Suppose that user decisions are noiseless (i.e., user decisions are given by (3)). For any $m \geq 2$, there exists a pure strategy equilibrium.*

### 3.1 Characterization of the equilibrium social loss for binary classification

We first focus on binary classification. Let $\mathcal{F}_{\text{all}}^{\text{binary}}$ denote the function class $\mathcal{F}_{\text{all}}^{\text{multi-class}}$ in the special case of $K = 2$ classes. Since $\mathcal{F}_{\text{all}}^{\text{binary}}$ lets model-providers make independent predictions about each representation $x$, the only source of error is noise in individual data points. To capture this, we define the *per-representation Bayes risk* $\alpha(x)$ to be $\alpha(x) := \min(\mathbb{P}(y = 1 \mid x), \mathbb{P}(y = 0 \mid x))$. The value $\alpha(x)$ measures how random the label $y$ is for a given representation $x$. As a result, $\alpha(x)$ is the minimum error that a model-provider can hope to achieve on the given representation $x$. Increasing $\alpha(x)$ increases the Bayes risk $\text{OPT}_{\text{single}}$: in particular, $\text{OPT}_{\text{single}}$ is equal to the average value $\mathbb{E}[\alpha(x)]$ across the population. The equilibrium social loss, however, depends on other aspects of $\alpha(x)$.

We characterize the equilibrium social loss in terms of the per-representation Bayes risks in the following proposition. Our characterization focuses on pure-strategy equilibria, which are guaranteed to exist in this setup (see Proposition 1).

**Proposition 2.** *Let $X$ be a finite set, let $K = 2$, and let $\mathcal{F} = \mathcal{F}_{\text{all}}^{\text{binary}}$. Suppose that user decisions are noiseless (i.e., user decisions are given by (3)). Suppose also that $\alpha(x) \neq 1/m$ for all $x \in X$.[3] At any pure strategy Nash equilibrium $f_1^*, \ldots, f_m^*$, the social loss $\text{SL}(f_1^*, \ldots, f_m^*)$ is equal to:*

$$\text{SL}(f_1^*, \ldots, f_m^*) = \mathbb{E}_{(x,y) \sim \mathcal{D}}\left[\alpha(x) \cdot \mathbb{1}[\alpha(x) < 1/m]\right]. \tag{4}$$

The primary driver of Proposition 2 is that as the per-representation Bayes risk $\alpha(x)$ decreases, the equilibrium predictions for $x$ go from *heterogeneous* (different model-providers offer different

---

[3]When $\alpha(x) = 1/m$, there turn out to be multiple pure-strategy equilibria with different social losses.

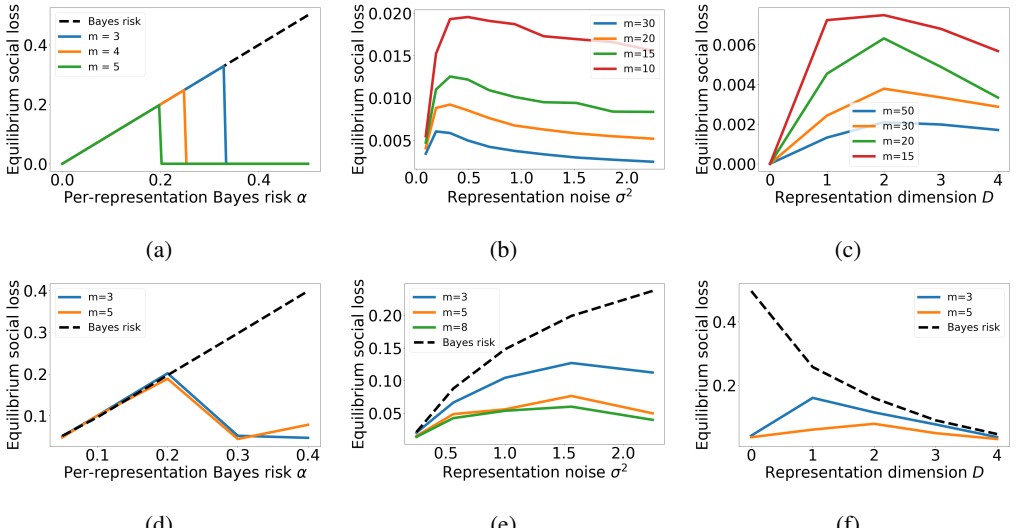

Figure 2: Equilibrium social loss (y-axis) versus data representation quality (x-axis) given $m$ model-providers, for different function classes $\mathcal{F}$ (rows) and when representations are varied along different aspects (columns). Top row: $\mathcal{F} = \mathcal{F}_{\text{all}}^{\text{binary}}$, with closed-form formula from Proposition 2. Bottom row: linear functions, computed via simulation (Section 4). We vary representations with respect to per-representation Bayes risk (a,d), noise level (b,e), and dimension (c,f). The dashed line indicates the Bayes risk (omitted if it is too high to fit on the axis). The Bayes risk is monotone, but the equilibrium social loss is non-monotone.

predictions for $x$) to *homogenous* (all model-providers offer the same prediction for $x$). In particular, if $\alpha(x)$ is below $1/m$, then all model-providers choose the Bayes optimal label $y^* = \arg\max_{y'} \mathbb{P}[y' \mid x]$, so predictions are homogeneous; on the other hand, if $\alpha(x)$ is above $1/m$, then at least one model-provider will choose $1 - y^*$, so predictions are heterogeneous. When predictions are heterogeneous, each user is offered perfect predictive accuracy by some model-provider, which results in zero social loss. On the other hand, if predictions are homogeneous and all model-providers choose the Bayes optimal label, the social loss on $x$ is the per-representation Bayes risk $\alpha(x)$. Putting this all together, the equilibrium social loss takes the value in (4). We defer a proof of Proposition 2 to Appendix D.

## 3.2 Non-monotonicity along several axes of varying representations

Using Proposition 2, we next vary representations along several axes and compute the equilibrium social loss, observing non-monotonicity in each case.

**Setting 1: Varying the per-representation Bayes risks.** Consider a population with a single value of $x$ with Bayes risk $\alpha(x) = \alpha$. We vary representation quality by varying $\alpha$ from 0 to 0.5. Figure 2a depicts the result: by Proposition 2, the equilibrium social loss is zero if $\alpha > 1/m$ and is $\alpha$ if $\alpha < 1/m$, leading to non-monotonicity at $\alpha = 1/m$. When there are $m \geq 3$ model-providers, the equilibrium social loss is thus non-monotonic in $\alpha$.[4] As $m$ increases, the non-monotonicity occurs at a higher data representation quality (a lower Bayes risk).

**Setting 2: Varying the representation noise.** Consider a one-dimensional population given by a mixture of two Gaussians (one for each class), where each Gaussian has variance $\sigma^2$ (see Appendix C for the details of the setup). We vary the parameter $\sigma$ to change the representation quality. Intuitively, a lower value of $\sigma$ makes the Gaussians more well-separated, which improves representation quality (Bayes risk). By Proposition 2, the equilibrium social loss is $\mathbb{E}[\alpha(x) \cdot \mathbb{1}[\alpha(x) < 1/m]]$. For each value of $\sigma$, we estimate the equilibrium social loss by sampling representations $x$ from the population and taking an average.[5] Figure 2b depicts the result: the equilibrium social loss is non-monotonic in $\sigma$

---

[4] For $m = 2$, where $\alpha = 1/2$ is the maximum possible per-representation Bayes risk, the equilibrium social loss is monotone in $\alpha$.

[5] Strictly speaking, we can't directly apply Proposition 2 to this setup since $X$ is infinite. We circumvent this issue by applying Proposition 2 on a sample of the representations.

(and thus the Bayes risk). Again, as the number of model-providers increases, the non-monotonicity occurs at a higher representation quality (a lower Bayes risk).

**Setting 3: Varying the representation dimension.** We consider a four-dimensional population $(X^{\text{all}}, Y)$, and let the representation $X$ consist of the first $D$ coordinates of $X^{\text{all}}$, for $D$ varying from 0 to 4 (see Appendix C for full details). Intuitively, a higher dimension $D$ makes the representations more informative, thus improving representation quality (Bayes risk). As before, for each value of $D$, we estimate the equilibrium social loss by sampling representations $x$ from the population and taking an average. Figure 2c depicts the result: the equilibrium social loss is once again non-monotonic in the representation dimension $D$ (and thus the Bayes risk).

**Discussion.** Settings 1-3 illustrate that equilibrium social loss can be non-monotonic in Bayes risk when representations are improved along many different axes. The intuition is that varying representations along these axes can increase $\alpha(x)$ for inputs $x$; by Proposition 2, these changes to $\alpha(x)$ can lead to non-monotonicity in the equilibrium social loss. We revisit Settings 1-3 for richer market structures (Appendix B.1) and for linear predictors and noisy user decisions (Section 4.2).

### 3.3 Generalization to more than 2 classes

While our analysis has thus far focused on classification with $K = 2$ classes, the number of classes $K$ can be much larger in practice. As a motivating example, consider content recommendation tasks where each class represents a different genre of content; since the content landscape can be quite diverse, we would expect $K$ to be fairly large. This motivates us to extend our theoretical characterization in Proposition 2 to classification with $K \geq 2$ classes.

For the case of $K \geq 2$ classes, the appropriate analogue of the per-representation Bayes risk is the per-class-per-representation Bayes risk, defined to be $\alpha^i(x) := \mathbb{P}(y = i \mid x)$ for each $x \in X$ and $i \in \{0, 1, \ldots, K - 1\}$. As a result, $1 - \max_{0 \leq i \leq K-1} \alpha^i(x)$ is the minimum error that a model-provider can achieve on $x$, and $\texttt{OPT}_{\text{single}}$ is equal to the average value $\mathbb{E}[1 - \max_{0 \leq i \leq K-1} \alpha^i(x)]$ across the population. The equilibrium social loss, however, depends on other aspects of the $\alpha^i(x)$ values.

We characterize the equilibrium social loss in terms of the per-class-per-representation Bayes risks in the following proposition. Our characterization again focuses on pure-strategy equilibria, which are guaranteed to exist in this setup by Proposition 1.

**Proposition 3.** *Let $X$ be a finite set, let $K \geq 2$, let $\mathcal{F} = \mathcal{F}_{all}^{multi-class}$. Suppose that user decisions are noiseless (i.e., user decisions are given by (3)). Let $c = \min_{x \in X} \max_{0 \leq i \leq K-1} \alpha^i(x)$. Then, at any pure strategy Nash equilibrium $f_1^*, \ldots, f_m^*$, the social loss $\texttt{SL}(f_1^*, \ldots, f_m^*)$ is bounded as*

$$
\mathbb{E}_{(x,y) \sim \mathcal{D}} \left[ \sum_{i=1}^{K} \alpha^i(x) \cdot \mathbb{1} \left[ \alpha^i(x) < \frac{c}{m} \right] \right] \leq \texttt{SL}(f_1^*, \ldots, f_m^*) \leq \mathbb{E}_{(x,y) \sim \mathcal{D}} \left[ \sum_{i=1}^{K} \alpha^i(x) \cdot \mathbb{1} \left[ \alpha^i(x) \leq \frac{1}{m} \right] \right].
\tag{5}
$$

The high-level intuition for Proposition 3 is similar to the intuition for Proposition 2, except that each class needs to be considered separately. In particular, when class $i$ occurs sufficiently frequently for the representation $x$ (i.e., when $\alpha^i(x)$ is not too small), then some model-provider will label $x$ as $i$; on the other hand, if the class $i$ occurs very infrequently for $x$, then no model-provider will label $x$ as $i$. We defer a proof of Proposition 3 to Appendix D.

While Proposition 3 is conceptually a generalization of Proposition 2, the details of Proposition 3 slightly differ. In particular, Proposition 3 does not completely pin down the equilibrium social loss, and there is a factor of $c$ slack in the constraint on each $\alpha^i(x)$ in (5) between the upper and lower bounds. Nonetheless, since the value $c = \min_{x \in X} \max_{0 \leq i \leq K-1} \alpha^i(x)$ measures the minimum accuracy of the Bayes optimal predictor across all inputs $x$, we expect that "reasonable" representations (i.e., sufficiently informative representations) would have $c$ equal to a constant. When $c$ is a constant, there is at most a constant factor slack in the $\alpha^i(x)$ constraints in (5) between upper and lower bound.

For similar reasons to Proposition 2, Proposition 3 implies that the equilibrium social loss can be non-monotonic in representation quality (i.e., Bayes risk). We defer a discussion to Appendix B.2.

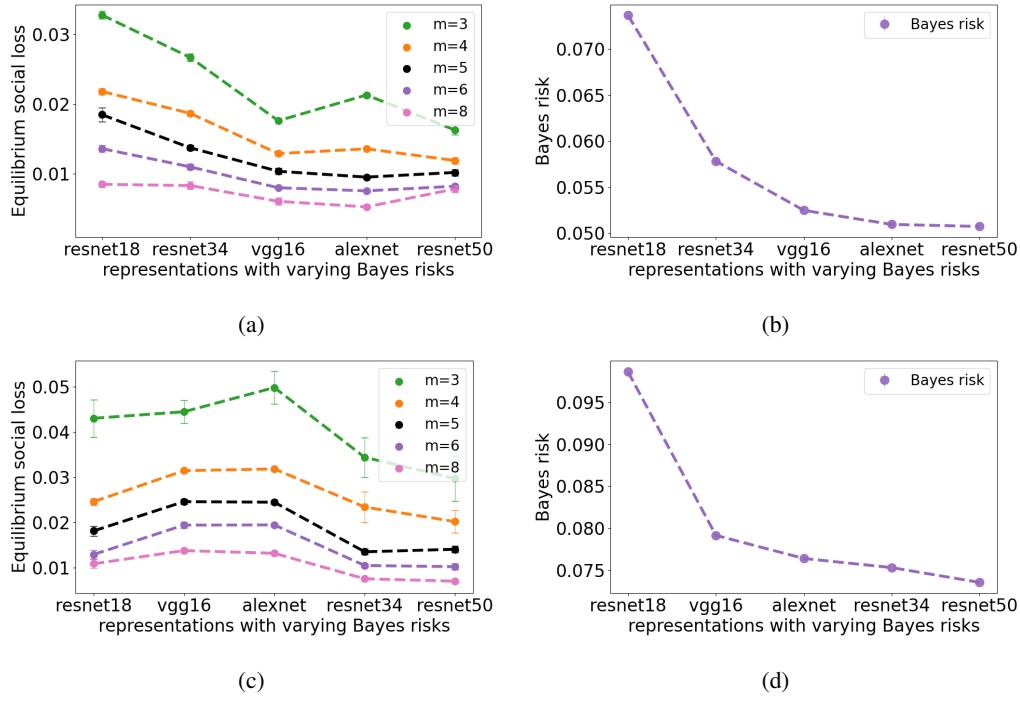

(a)

(b)

(c)

(d)

Figure 3: Equilibrium social loss (left) and Bayes risk (right) on a binary (top) and multi-class (bottom) classification tasks on CIFAR-10 (Section 4.3). Representations are generated from different networks pretrained on ImageNet. The points show the equilibrium social loss when $m$ model-providers compete with each other (left) and the Bayes risk of a single model-provider in isolation (right). While Bayes risk is decreasing in this representation ordering, the equilibrium social loss is non-decreasing in this ordering. The equilibrium social loss is thus non-monotonic in representation quality as measured by Bayes risk. Error bars are 1 standard error.

## 4 Empirical Analysis of Non-monotonicity for Linear Predictors

We next turn to linear predictors and demonstrate empirically that the social welfare can be non-monotonic in data representation quality in this setup as well. We take $X = \mathbb{R}^D$ and we let the model parameters be $\phi$. For binary classification, we let $\mathcal{F}_{\text{linear}}^{\text{binary}}$ be the family of linear predictors $f_{w,b} = \text{sigmoid}(\langle w, x \rangle + b)$ where $w \in \mathbb{R}^D$, $b \in \mathbb{R}$, and $\phi = [w, b]$. Similarly, for classification with more than 2 classes, we let $\mathcal{F}_{\text{linear}}^{\text{multi-class}}$ be the family of linear predictors $f_{W,b} = \text{softmax}(Wx + b)$ where $w \in \mathbb{R}^{|Y| \times D}$, $b \in \mathbb{R}^{|Y|}$, and $\phi = [W, b]$. Since this setting no longer admits closed-form formulae, we numerically estimate the equilibria using a variant of *best-response dynamics*, where model-providers repeatedly best-respond to the other predictors.

We first show on low-dimensional synthetic data on a binary classification task that the insights from Section 3.2 generalize to linear predictors (Figures 2d-2f). We then turn to natural data, considering binary and 10-class image classification tasks for CIFAR-10 and using pretrained networks—AlexNet, VGG16, and various ResNets—to generate high-dimensional representations (ranging from 512 to 4096). We again find that the equilibrium social loss can be non-monotonic in Bayes risk (Figure 3).

### 4.1 Best-response dynamics implementation

To enable efficient computation, we assume the distribution $\mathcal{D}$ corresponds to a finite dataset with $N$ data points. We calculate equilibria using an approximation of best-response dynamics. Model-providers (players) iteratively (and approximately) best-respond to the other players' actions. We implement the approximate best-response as running several steps of gradient descent.

In more detail, for each $j \in [m]$, we initialize the model parameters $\phi$ as mean zero Gaussians with standard deviation $\sigma$. Our algorithm then proceeds in stages. At a given stage, we iterate through the model-providers in the order $1, \ldots, m$. When $j$ is chosen, first we decide whether to reinitialize:

if the risk $\mathbb{E}_{(x,y)\sim\mathcal{D}}[\ell(f_\phi(x), y)]$ exceeds a threshold $\rho$, we re-initialize $w_j$ and $b_j$ (sampling from mean zero Gaussians as before); otherwise, we do not reinitialize. Then we run gradient descent on $u(\cdot; \mathbf{f}_{-j})$ (computing the gradient on the full dataset of $N$ points) with learning rate $\eta$ for $I$ iterations, updating the parameters $\phi$. We run this gradient descent step up to 2 more times if the risk $\mathbb{E}_{(x,y)\sim\mathcal{D}}[\ell(f_\phi(x), y)]$ exceeds a threshold $\rho'$. At the end of a stage, the stopping condition is that for every $j \in [m]$, model-provider $j$'s utility $u(f_j, \mathbf{f}_{-j})$ has changed by at most $\epsilon$ relative to the previous stage. If the stopping condition is not met, we proceed to the next stage.[6]

## 4.2 Simulations on synthetic data

We first revisit Settings 1-3 from Section 3.2, considering the same axes of varying representations and distributions over $(x, y)$. In contrast to Section 3.2, we restrict the model family to linear predictors $\mathcal{F}_{\text{linear}}^{\text{binary}}$ instead of allowing all predictors $\mathcal{F}_{\text{all}}^{\text{binary}}$. We also set the noise parameter $c$ in user decisions (1) to 0.3. Our goal is to examine if the findings from Section 3 generalize to this new setting.

We compute the equilibria for each of the following (continuous) distributions as follows. First, we let $\mathcal{D}$ be the empirical distribution over $N = 10,000$ samples from the continuous distribution. Then we run the best-response dynamics described in Section 4.1 with $\rho = \rho' = 1$, $I = 5000$, $\eta = 0.1$, $\epsilon = 0.001$, and $\sigma = 1.0$. We then compute the equilibrium social loss according to (2). We also compute the Bayes optimal predictor with gradient descent. See Appendix C for full details.

Our results, described below, are depicted in Figures 2d-2f (row 2). We compare these results with Figures 2a-2c (row 1), which shows the analogous results for $\mathcal{F}_{\text{all}}^{\text{binary}}$ from Section 3.2.

**Setting 1: Varying the per-representation Bayes risks.** Consider the same single $x$ setup as in Setting 1 in Section 3.2. The only parameter of the predictor is the bias $b \in \mathbb{R}$ (i.e., we treat $x$ as zero-dimensional). Figure 2d shows that the equilibrium social loss is non-monotonic in $\alpha$, which mirrors the non-monotonicity in Figure 2a.

**Setting 2: Varying the representation noise.** Consider the same one-dimensional mixture-of-Gaussians distribution as in Setting 2 in Section 3.2. (The weight $w$ is one-dimensional.) We again vary the noise $\sigma$ to change the representation quality. Figure 2e shows that the equilibrium social loss is non-monotonic in the noise $\sigma$, which again mirrors the non-monotonicity in Figure 2b.

**Setting 3: Varying the representation dimension.** Consider the same four-dimensional population as in Setting 3 in Section 3.2. We vary the representation dimension $D$ from 0 to 4 to change the representation quality. Figure 2f shows that the equilibrium social loss is non-monotonic in the dimension $D$, which once again mirrors the non-monotonicity in Figure 2c.

**Discussion.** In summary, in Figure 2, rows 1 and 2 exhibit similar non-monotonicities. This illustrates that the insights from Section 3.1 translate to linear predictors and noisy user decisions.

## 4.3 Simulations on CIFAR-10 for binary classification

We next turn to experiments with natural data. While we have directly varied the informativeness of data representations thus far, representations in practice are frequently generated by pretrained models. The choice of the pretrained model implicitly influences representation quality, as measured by Bayes risk on the downstream task. In this section, we consider how the equilibrium social loss changes with representations generated from pretrained models of varying quality. We restrict the model family to linear predictors $\mathcal{F}_{\text{linear}}^{\text{binary}}$ and set the noise parameter $c$ in user decisions (1) to 0.1.

We consider a binary image classification task on CIFAR-10 [Krizhevsky, 2009] with 50,000 images. Class 0 is defined to be $\{\text{airplane}, \text{bird}, \text{automobile}, \text{ship}, \text{horse}, \text{truck}\}$ and the class 1 is defined to be $\{\text{cat}, \text{deer}, \text{dog}, \text{frog}\}$. We treat the set of 50,000 images and labels as the population of users, meaning that it is both the training set and the validation set.[7] Representations are generated from five models—AlexNet [Krizhevsky et al., 2012], VGG16 [Simonyan and Zisserman, 2015], ResNet18, ResNet34, and ResNet50 [He et al., 2016]—pretrained on ImageNet [Deng et al., 2009]. The

---

[6]The code can be found at https://github.com/mjagadeesan/competition-nonmonotonicity.

[7]We make this choice to be consistent with the rest of the paper, where we focus on population-level behavior and thus do not consider generalization error.

representation dimension is 4096 for AlexNet and VGG16, 512 for ResNet18 and ResNet34, and 2048 for ResNet50.

We compute the equilibria as follows. First, let $\mathcal{D}$ be the distribution described above with $N = 50,000$ data points. Then we run the best-response dynamics described in Section 4.1 for $m \in \{3, 4, 5, 6, 8\}$ model-providers with $\rho = \rho' = 0.3$, $I = 2000$, $\epsilon = 0.001$, $\sigma = 0.5$, and a learning rate schedule that starts at $\eta = 1.0$. We then compute the equilibrium social loss according to (2). We also compute the Bayes risk using gradient descent. For experimental details, see Appendix C.

Figures 3a-3b show that the equilibrium social loss can be non-monotone in the Bayes risk. For example, for $m = 3$, VGG16 outperforms AlexNet, even though the Bayes risk of VGG16 is much higher than the Bayes risk of AlexNet. Interestingly, the location of the non-monotonicity differs across different values of $m$. For example, for $m = 5$ and $m = 8$, AlexNet outperforms ResNet50 despite having a higher Bayes risk, but ResNet50 outperforms AlexNet for $m = 3$ and $m = 4$.

### 4.4 Simulations on CIFAR-10 for 10-class classification

While our empirical analysis has thus far focused on binary classification, we now turn to classification with more than 2 classes. In particular, we consider a ten class CIFAR-10 [Krizhevsky, 2009] task with 50,000 images. The labels are specified by the CIFAR-10 classes in the original dataset. We treat the set of 50,000 images and labels as the population of users, meaning that it is both the training set and the validation set. Representations are generated from the same five models as in Section 4.3. We restrict to linear predictors $\mathcal{F}_{\text{linear}}^{\text{multi-class}}$ and again set the noise parameter $c$ in user decisions (1) to 0.1.

We compute the equilibria. We again let $\mathcal{D}$ be the distribution described above with $N = 50,000$ data points. Then, we run the best-response dynamics described in Section 4.1 for $m \in \{3, 4, 5, 6, 8\}$, and we compute the equilibria with the same hyperparameter settings as in Section 4.3 (except that $\rho = 0.7$ and $\rho' = 1$). We then compute the equilibrium social loss according to (2). We also compute the Bayes risk using gradient descent. For full experimental details, see Appendix C.

Figures 3c-3d show that the equilibrium social loss can be non-monotone in the Bayes risk. For example, across all five values of $m$, ResNet18 outperforms VGG16, even though the Bayes risk of ResNet is substantially higher than the Bayes risk of VGG16. Furthermore, for $m = 3$, VGG16 outperforms AlexNet despite having a larger Bayes risk. Interestingly, the shape of the equilibrium social loss curve for each value of $m$ (Figure 3c) appears qualitatively different than the analogous equilibrium social loss curve for binary classification (Figure 3a).

## 5 Discussion

We showed that the monotonicity of scaling trends can be violated under competition. We demonstrated that when multiple model-providers compete for users, improving data representation quality (as measured by Bayes risk) can *increase* overall loss at equilibrium. We exhibited non-monotonicity of the equilibrium social loss in Bayes risk when representations are varied along several axes (per-representation Bayes risk, noise, dimension, and pre-trained model which generates representations).

An interesting direction for future work is to further characterize the regimes when the equilibrium social loss is monotonic versus non-monotonic in data representation quality as measured by Bayes risk. For example, it would be interesting to generalize our theoretical results from Section 3 to general function classes and market reputations as well as to generalize our empirical findings from Section 4 to other axes of varying data representations and to non-linear functions. Finally, while we focused on classification, an interesting direction would be to generalize our findings to regression tasks with continuous outputs or to generative AI tasks with text-based or image-based outputs.

More broadly, the non-monotonicity of equilibrium social welfare in scale under competition establishes a disconnect between scaling trends in the single model-provider setting and in the competitive setting. In particular, typical scaling trends (e.g. [Kaplan et al., 2020, Sharma and Kaplan, 2020, Bahri et al., 2021, Hoffmann et al., 2022, Hernandez et al., 2021]) may not translate to competitive settings such as digital marketplaces. Thus, understanding the downstream impact of scale on user welfare in digital marketplaces will likely require understanding how scaling trends behave under competition. We hope that our work serves as a starting point for analyzing and eventually characterizing the scaling trends of learning systems in competitive settings.

# 6 Acknowledgments

We thank Yiding Feng, Xinyan Hu, and Alex Wei for useful comments on the paper. This work was partially supported by the Open Phil AI Fellowship, the Berkeley Fellowship, the European Research Council (ERC) Synergy Grant program, and the National Science Foundation under grant numbers 2031899 and 1804794.

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
