# A Discussion of Model Assumptions

We highlight and discuss several assumptions that we make in our stylized model.

## A.1 Assumptions on user decisions

Our primary model for user decisions given by (1) is the standard logit model for discrete choice decisions [Train, 2002] which is also known as the Boltzmann rationality model. In the limit as $c \to 0$, a user with representation $x$ and label $y$ select from the set of model-providers $\arg\min_{j \in [m]} \ell(f_j(x), y)$ that achieve the minimum loss; in particular, the user chooses a model-provider from this set with probability proportional to the model-provider's market reputation. For $c > 0$, the specification in equation (1) captures that users evaluate a model-provider based on a noisy perception of the loss.

While this model implicitly assumes that a user's choice of platform is fully specified by the platforms' choices of predictor (i.e. platforms are ex-ante homogeneous), we extend this model in Appendix B.1 to account for uneven market reputations across decisions. These market reputations are modeled as global weights in the logit model for discrete choice. Given market reputations $w_1, \ldots, w_m$, users choose a predictor according to:

$$\mathbb{P}[j^*(x, y) = j] = \frac{w_j \cdot e^{-\ell(f_j(x), y)/c}}{\sum_{j'=1}^m w_{j'} \cdot e^{-\ell(f_{j'}(x), y)/c}}. \tag{6}$$

When the market reputations are all equal ($w_1 = \ldots = w_m$), equation (6) exactly corresponds to (1). When the market reputations $w_j$ are not equal, equation (6) captures that users place a higher weight on model-providers with a higher market reputation. This captures that users are more likely to choose a popular model-provider than a very small model-provider without much reputation. However, this formalization does assume that market reputations are global across users and that market reputations surface as tie-breaking weight in the noiseless limit.

Implicit in this model is asymmetric information between the model-providers and users. While the only information that a model-provider has about users is their representations, a user can make decisions based on noisy perceptions of their own loss (which can depend on their label). This captures that, even if users are unlikely to know their own labels, users can experiment with multiple model-providers to (noisily) determine which one maximizes their utility. The inclusion of market reputations reflects that users are more likely to experiment with and ultimately choose popular model-providers than less popular model-providers.

## A.2 Assumption of global data representations

Our results assume that all model-providers share the same representations $x$ for each user and thus improvements in representations $x$ are experienced by all model-providers. This assumption is motivated by emerging marketplaces where different model-providers utilize the same pretrained model, but *finetune* the model in different ways. To simplify this complex training process, we conceptualize pretraining as *learning data representations (e.g., features)* and fine-tuning as *learning a predictor from these representations*. In this formalization, increasing the scale of the pretrained model (e.g., by increasing the number of parameters or the amount of data) leads to improvements in data representations accessible to all of the model-providers during "fine-tuning".

An interesting direction for future work would be to incorporate heterogeneity or local improvements in the data representations.

## A.3 Assumption on model-provider action space

We make the simplifying assumption that the only action taken by model-providers is to choose a classifier from a pre-specified class. This formalization does not capture other actions (such as data collection and price setting) that may be taken by the platform. Incorporating other model-provider decisions would be an interesting avenue for future work.

# B Additional results for Section 3

## B.1 Generalization to unequal market reputations

While we assumed above that users evenly break ties between model-providers, in reality, users might be more likely to choose model-providers with a higher market reputation (e.g., established, popular model-providers). This motivates us to incorporate market reputations into user decisions.

Formally, we assign to each model-provider $j$ a *market reputation* $w_j$, and we replace the logit model in (1) with a weighted logit variant. When $c \to 0$, rather than breaking ties uniformly, they are instead broken proportionally to $w_j$:

$$\mathbb{P}[j^*(x,y) = j] = \begin{cases} 0 & \text{if } j \notin \arg\min_{j' \in [m]} \mathbb{1}[y \neq f_{j'}(x)] \\ \frac{w_j}{\sum_{j'' \in [m]} w_{j''} \cdot \mathbb{1}[j'' \in \arg\min_{j' \in [m]} \mathbb{1}[y \neq f_{j'}(x)]]} & \text{if } j \in \arg\min_{j' \in [m]} \mathbb{1}[y \neq f_{j'}(x)]. \end{cases}$$

(7)

See Appendix A for further discussion of this model. For simplicity, we assume that market reputations are normalized to sum to one.

Similarly to Proposition 2, we derive a closed-form formula for the equilibrium social loss, focusing on the case of binary classification with $m = 2$ model-providers for analytic tractability. We observe non-monotonicity as before, but with a more complex functional form.

**Proposition 4.** *Let $X$ be a finite set, let $K = 2$, and let $\mathcal{F} = \mathcal{F}_{all}^{binary}$. Suppose there are $m = 2$ model-providers with market reputations $w_{min}$ and $w_{max}$, where $w_{max} \geq w_{min}$ and $w_{max} + w_{min} = 1$. Suppose that user decisions are given by (7), and that $\alpha(x) \neq w_{min}$ for all $x \in X$.[8] At any (mixed) Nash equilibrium $(\mu_1, \mu_2)$, the expected social loss is equal to:*

$$\mathbb{E}_{\substack{f_1 \sim \mu_1 \\ f_2 \sim \mu_2}}[SL(f_1, f_2)] = \mathbb{E}_{(x,y) \sim \mathcal{D}}\left[\underbrace{\frac{(\alpha(x) - w_{min}) \cdot (w_{max} - \alpha(x))}{(1 - 2 \cdot w_{min})^2}}_{(A)} \cdot \mathbb{1}[\alpha(x) > w_{min}] + \underbrace{\alpha(x)}_{(B)} \cdot \mathbb{1}[\alpha(x) < w_{min}]\right].$$

(8)

The high-level intuition for Proposition 4, like for Proposition 2, is that the equilibrium predictions go from heterogeneous to homogenous as $\alpha(x)$ decreases. Term (A), which is realized for large $\alpha(x)$, captures the equilibrium social loss for heterogeneous predictions. Term (B), which is realized for small $\alpha(x)$, captures the equilibrium social loss for homogeneous predictions. We defer the proof of Proposition 4 to Appendix D.

The details of Proposition 4 differ from Proposition 2 in several ways. First, the transition point from heterogeneous to homogeneous predictions occurs at $\alpha(x) = w_{min}$ as opposed to $\alpha(x) = 1/2$. In particular, the transition point depends on the market reputations rather than only the number of model-providers. Second, the equilibria have *mixed strategies* rather than pure strategies, because pure-strategy equilibria do not necessarily exist when market reputations are unequal (see Lemma 7 in Appendix D). Third, the social loss at a representation $x$ is no longer equal to zero for heterogeneous predictions—in particular, term (A) is now positive for all $\alpha(x) > w_{min}$ and increasing in $\alpha(x)$.

To better understand the implications of Proposition 4, we revisit Settings 1-3 from Section 3.2, considering the same three axes of varying representations with the same distributions over $(x, y)$. In contrast to Section 3.2, we consider 2 competing model-providers with unequal market positions rather than $m$ competing model providers with equal market positions. Our results, described below, are depicted in Figure 4.

**Setting 1: Varying the per-representation Bayes risks.** Consider the same setup as Setting 1 in Section 3.2. Figure 4a depicts the non-monotonicity of the equilibrium social loss in the per-representation Bayes risk $\alpha$ across different settings of market reputations for 2 competing model-providers. The discontinuity occurs at the smaller market reputation $w_{min}$. Thus, as the market reputations of the 2 model-providers become closer together, the non-monotonicity occurs at a lower data representation quality (higher Bayes risk).

---

[8]As with Proposition 2, when $\alpha(x)$ is equal to $w_{min}$ for some value of $x$, there are multiple equilibria.

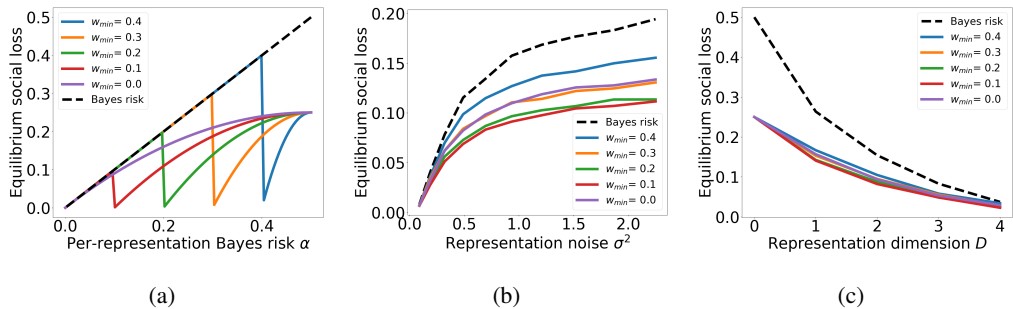

Figure 4: Equilibrium social loss (y-axis) versus data representation quality (x-axis) given two model-providers with market reputations $[1 - w_{\min}, w_{\min}]$ when representations are varied along different aspects (columns). The equilibrium social loss is computed via the closed-form formula from Proposition 4. We vary representations with respect to per-representation Bayes risk (a), noise level (b), and dimension (c). The dashed line indicates the Bayes risk. The Bayes risk is monotone for all 3 axes of varying representations; on the other hand, the equilibrium social loss is non-monotone in the per-representation Bayes risk and monotone in noise level and dimension.

**Settings 2-3: Varying the representation noise or representation dimension.** Consider the setups from Settings 2-3 in Section 3.2. Figures 4b-4c depicts that the equilibrium social loss is *monotone* in data representation quality (Bayes risk) across different settings of market reputations for 2 competing model-providers.

**Discussion.** To interpret these results, observe that for 2 model-providers with equal market reputations ($w_{\min} = 0.5$), the equilibrium social loss is always equal to the Bayes risk by Propositions 2-4, which trivially implies monotonicity. In contrast, Figure 4 shows that for unequal market positions ($w_{\min} < 0.5$), the equilibrium social loss is non-monotonic in Bayes risk for Setting 1, though it is still monotonic in Bayes risk for Settings 2 and 3. (For comparison, recall from Figures 2a-2c that for $m \gg 2$ model-providers with equal market reputations, non-monotonicity was exhibited for all three settings.) An interesting open question is identify other axes of varying representations, beyond Setting 1, which lead to non-monotonicity for 2 model-providers with unequal market reputations.

### B.2 Additional example for Section 3.3

For similar reasons to Proposition 2, Proposition 3 implies that the equilibrium social loss can be non-monotonic in the representation quality (i.e., the Bayes risk). As a concrete example, consider the following generalization of Setting 1 in Section 3.1: let there be a population with a single value of $x$ where $\alpha_0(x) = 1 - 2\alpha$, $\alpha_1(x) = \alpha$, and $\alpha_2(x) = \alpha$ for $\alpha < 1/4$. In this case, we see that $c \geq 1/2$ for any $\alpha < 1/4$. By Proposition 3, the equilibrium social loss is nonzero is $\alpha < 1/(2m)$, but is zero if $\alpha > 1/m$, whereas the Bayes risk is equal to $2\alpha$, which shows non-monotonicity as desired. We expect that other setups similar to those in Section 3.1 will also lead to non-monotonicity for multi-class tasks.

## C Additional Details of Simulations

**Hyperparameters.** We introduce a temperature parameter $\tau$ within our loss function, defining the loss $\ell(f_{w,b}(x), y)$ to be $|\operatorname{sigmoid}((\langle w, x \rangle + b)/\tau) - 1|$. This reparameterizes, but does not change, the model family.

When we run the best-response dynamics, we always initialize the model parameters as mean-zero Gaussians with standard deviation $\sigma$. When we reinitialize model parameters, we again initialize them as mean-zero Gaussians with standard deviation $\sigma$. For Section 4.2, we set $I = 5000$, $\tau = 0.1$, $\epsilon = 0.001$, $\eta = 0.1$, $\sigma = 1.0$, and $\rho = \rho' = 1.0$. For Section 4.3 and Section 4.4, we set $I = 2000$, $\sigma = 0.5$, $\tau = 1.0$, $\epsilon = 0.001$, and $\eta$ with the following learning rate schedule to expedite convergence: $\eta = 1.0$ if the risk $\mathbb{E}_{(x,y)\sim\mathcal{D}}[\ell(f_{w_j,b_j}(x), y)]$ is at least 0.5, $\eta = 5.0$ if the risk is in $[0.4, 0.5)$, $\eta = 15$ if the risk is in $[0.3, 0.4)$, and $\eta = 20$ if the risk is less than 0.3. We set $\rho = \rho' = 0.3$ for Section 4.3 and we set $\rho = 0.7$ and $\rho' = 1$ for Section 4.3.

For Section [4.3] and Section [4.4], we ran over several trials for each data point and the error bars show two standard errors from the mean. For binary classification, the number of trials was 20 for $m = 3$ and $m = 4$ and 8 for $m = 5$, $m = 6$, and $m = 8$. For 10-class classification, the number of trials was 40 for $m = 3$ and $m = 4$ and 8 for $m = 5$, $m = 6$, and $m = 8$.

In addition to computing the equilibria, we also approximate the optimal Bayes risk. For Section [4.2], we run gradient descent for $10,000$ iterations with learning rate equal to one and parameters initialized to independent Gaussians with zero mean and standard deviation 1.0. For Section [4.3], we run gradient descent for $50,000$ iterations with learning rate equal to 0.1 and parameters initialized to independent Gaussians with zero mean and standard deviation 0.005. For Section [4.4], we run gradient descent for $70,000$ iterations with learning rate equal to 0.1 and parameters initialized to independent Gaussians with zero mean and standard deviation 0.005.

**Generation of the synthetic dataset.** In Setting 1 (Figures [2a], [4a], and [2d]), we consider a zero-dimensional population where $Y \mid X$ is distributed as a Bernoulli with probability $\alpha$. In Figure [2d], the meaning of a zero-dimensional representation is that the only parameter is the bias.

In Setting 2 (Figures [2b], [4b], and [2e]), we consider a one-dimensional population given by a mixture of Gaussians. In particular, the Gaussian $X \mid Y = 0$ is distributed as $N(-\mu, \sigma^2)$ and the Gaussian $X \mid Y = 1$ is distributed as $N(\mu, \sigma^2)$. The mean $\mu$ is taken to be 1. The distribution of the labels is given by $\mathbb{P}[Y = 1] = 0.4$ and $\mathbb{P}[Y = 1] = 0.6$.

In Setting 3 (Figures [2c], [4c], and [2f]), let $D_{\text{base}} = 4$. The distribution over $(X^{\text{all}}, Y)$ consists of $D_{\text{base}}$ subpopulations. We define the distribution of $(X^{\text{all}}, Y)$ as follows: each subpopulation $1 \le i \le D_{\text{base}}$ has a different mean vector $\mu_i \in \mathbb{R}^{D_{\text{base}}}$ and is distributed as $X^{\text{all}} \sim Y = 0 \sim N(-\mu_i, \sigma^2)$, let $X^{\text{all}} \sim Y = 1 \sim N(\mu_i, \sigma^2)$, and let $\mathbb{P}[Y = 0] = \mathbb{P}[Y = 1] = 1/2$. We define $(\mu_i)_d = 0$ for $1 \le d \le i - 1$ and $(\mu_i)_d = 1$ for $i \le d \le D_{\text{base}}$, and we let $\sigma = 1$. If the representation dimension is $D$, then we define $X$ to consist of the first $D$ coordinates of $X^{\text{all}}$. When $D = 0$, the model-provider is not given representations and thus must assign all users to the same output. (Our setup captures that the dimension $D$ must be at least $i$ to see any nontrivial features about subpopulation $i$.) The distribution across the 4 subpopulations is 0.7, 0.15, 0.1, and 0.05.

In each case, we draw 10,000 samples and take the resulting empirical distribution to be $\mathcal{D}$.

**Generation of the CIFAR-10 task.** We consider a binary classification task consisting of the first 10,000 images in the training set of CIFAR-10. The class 0 is defined to be {airplane, bird, automobile, ship, horse, truck} and class 1 is defined to be {cat, deer, dog, frog}. To generate representations, we use the pretrained models from the Pytorch `torchvision.models` package; these models were pretrained on ImageNet.

**Compute details.** We run our simulations on a single A100 GPU.

# D Additional Results and Proofs for Section [3]

In Appendix [D.1], we show a decomposition lemma and prove existence of equilibrium (Proposition [1]). We prove the results from Section [3.1] in Appendix [D.2], prove the results from Section [3.3] in Appendix [D.3], and prove the results from Section [B.1] in Appendix [D.4].

## D.1 Decomposition lemma and existence of equilibrium

We first show that we can decompose model-provider actions into independent decisions about each representation $x$. To formalize this, let $\mathcal{D}$ be the data distribution, and let $\mathcal{D}_x$ be the conditional distribution over $(X, Y) \mid X = x$ where $(X, Y) \sim \mathcal{D}$. Let $(\mathcal{F}_{\text{all}}^{\text{multi-class}})^x := \{f^0, f^1, \ldots, f^{K-1}\}$ be the class of $K$ functions from a single representation $x$ to $\{0, 1, \ldots, K-1\}$, where $f^i(x) = i$.

**Lemma 5.** *Let $X$ be a finite set of representations, let $\mathcal{F} = \mathcal{F}_{\text{all}}^{\text{multi-class}}$, and let $\mathcal{D}$ be the distribution over $(X, Y)$. For each $x \in X$, let $\mathcal{D}_x$ be the conditional distribution over $(X, Y) \mid X = x$ where $(X, Y) \sim \mathcal{D}$, and let $(\mathcal{F}_{\text{all}}^{\text{multi-class}})^x := \{f^0, f^1, \ldots, f^{K-1}\}$ be the class of the $K$ functions from a single representation $x$ to $\{0, 1\}$, where $f^i(x) = i$. Suppose that user decisions are noiseless (i.e., $c \to 0$, so user decisions are given by [(3)]). A market outcome $f_1, \ldots, f_m$ is a pure-strategy equilibrium if and only if for every $x \in X$, the market outcome $(f^{f_1(x)}, \ldots, f^{f_m(x)})$ is a pure-strategy equilibrium for $(\mathcal{F}_{\text{all}}^{\text{multi-class}})^x$ with data distribution $\mathcal{D}_x$.*

The intuition is that since $\mathcal{F}_{\text{all}}^{\text{multi-class}}$ is all possible functions, model-providers make independent decisions for each data representation.

*Proof.* Let $\mathcal{D}^R$ be the marginal distribution of $X$ with respect to the distribution $(X, Y) \sim \mathcal{D}$. First, we write model-provider $j$'s utility as:

$$u(f_j; \mathbf{f}_{-j}) = \mathop{\mathbb{E}}_{(x,y)\sim\mathcal{D}} [\mathbb{P}[j^*(x, y) = j]] = \mathop{\mathbb{E}}_{x'\sim\mathcal{D}^R} \left[ \mathop{\mathbb{E}}_{(x,y)\sim\mathcal{D}_{x'}} [\mathbb{P}[j^*(x, y) = j]] \right], \quad (9)$$

where $\mathbf{f}_{-j}$ denotes the predictors chosen by the other model-providers. The key intuition for the proof will be that the predictions $[f_1(x''), \ldots, f_m(x'')]$ affect $\mathbb{E}_{(x,y)\sim\mathcal{D}_{x'}} [\mathbb{P}[j^*(x, y) = j]]$ if and only if $x' = x''$.

First we show that if $f_1, \ldots, f_m$ is a pure-strategy equilibrium, then $(f^{f_1(x')}, \ldots, f^{f_m(x')})$ is a pure-strategy equilibrium for $(\mathcal{F}_{\text{all}}^{\text{multi-class}})^{x'}$ with data distribution $\mathcal{D}_{x'}$. Assume for sake of contradiction that $(f^{f_1(x')}, \ldots, f^{f_m(x')})$ is not an equilibrium. Then there exists $j' \in [m]$ such that model-provider $j'$ would achieve higher utility if they switched from $f^{f_{j'}(x')}$ to $f^l$ for some $l \neq f_{j'}(x')$. Let $f'_{j'}$ be the predictor given by $f'_{j'}(x) = f_{j'}(x)$ if $x \neq x'$ and $f'_{j'}(x') = l$. By equation (9), this would mean that $u(f'_{j'}; \mathbf{f}_{-j'})$ is strictly higher than $u(f_{j'}; \mathbf{f}_{-j'})$ which is a contradiction.

Next, we show that if $(f^{f_1(x')}, \ldots, f^{f_m(x')})$ is a pure-strategy equilibrium for $(\mathcal{F}_{\text{all}}^{\text{binary}})^{x'}$ with data distribution $\mathcal{D}_{x'}$ for all $x' \in X$ then $f_1, \ldots, f_m$ is a pure-strategy equilibrium. Assume for sake of contradiction that there exists $j'$ such that $u(f'_{j'}; \mathbf{f}_{-j'}) > u(f_j; \mathbf{f}_{-j'})$. By equation (9), there must exist $x'$ such that $\mathbb{E}_{(x,y)\sim\mathcal{D}_{x'}} [\mathbb{P}[j^*(x, y) = j']]$ is higher for $f'_{j'}$ than $f_{j'}$. This means that $(f^{f_1(x')}, \ldots, f^{f_m(x')})$ is not an equilibrium (since $f^l$ would be a better response for model-provider $j'$) which is a contradiction. $\square$

We next prove Proposition 1, showing that a pure-strategy equilibrium exists by applying the proof technique of Lemma 3.7 of Ben-Porat and Tennenholtz [2019].

*Proof of Proposition 1.* By Lemma 5, it suffices to show that there exists a pure-strategy equilibrium whenever there is a single data representation $X = \{x\}$. In this case, the function class $\mathcal{F}_{\text{all}}^{\text{multi-class}}$ consists of $K$ predictors $\{f^0, f^1 \ldots, f^{K-1}\}$ given by $f^i(x) = i$. For each class $i$, let $\mathbb{P}[Y = i \mid X] = p_i$.

For the special case of $K = 2$ (binary classification), the game between model-providers is thus a 2-action game with symmetric utility functions. Thus, it must possess a (possibly asymmetric) pure Nash equilibrium [Cheng et al., 2004].

For the general case of $K \geq 2$, we can no longer apply the result in [Cheng et al., 2004] since there can be more than 2 actions. We instead show that the game is a potential game, following a similar argument to Ben-Porat and Tennenholtz [2019]. We define the potential function $\Phi(\cdot)$ as follows. For each $i \in \{0, 1, \ldots, K-1\}$, we define the function $G_i : \{f^0, f^1 \ldots, f^{K-1}\}^m \to \mathbb{R}_{\geq 0}$ to be:

$$G_i(f_1, \ldots, f_m) := \begin{cases} \frac{1}{m} & \text{if } |\{j \in [m] \mid f_j = f^i\}| = 0 \\ \sum_{l=1}^{|\{j\in[m]|f_j=f^i\}|} \frac{1}{l} & \text{if } |\{j \in [m] \mid f_j = f^i\}| \geq 1. \end{cases}$$

We let

$$\Phi(f_1, \ldots, f_m) := \sum_{i=1}^{K} p_i \cdot G_i(f_1, \ldots, f_m).$$

We show that $\Phi$ is a potential function for this game. Suppose that model-provider $j$ switches from $f_j := f^i$ to $f'_j = f^{i'}$ for $i' \neq i$. For each $i \in \{0, 1, \ldots, K-1\}$, let $N_i = |\{j \in [m] \mid f_j = f^i\}|$ be the number of model-providers who choose $f^i$ on the original outcome $[f_1, \ldots, f_m]$. We observe that:

$$u(f_j; \mathbf{f}_{-j}) - u(f'_j; \mathbf{f}_{-j}) = \begin{cases} p_i \cdot \frac{1}{N_i} - p_{i'} \cdot \frac{1}{N_{i'}+1} & \text{if } N_i > 1, N_{i'} > 0 \\ p_i \cdot \left(1 - \frac{1}{m}\right) - p_{i'} \cdot \frac{1}{N_{i'}+1} & \text{if } N_i = 1, N_{i'} > 0 \\ p_i \cdot \frac{1}{N_i} - p_{i'} \cdot \left(1 - \frac{1}{m}\right) & \text{if } N_i > 1, N_{i'} = 0 \\ p_i \cdot \left(1 - \frac{1}{m}\right) - p_{i'} \cdot \left(1 - \frac{1}{m}\right) & \text{if } N_i = 1, N_{i'} = 0 \end{cases}$$

Moreover, we see that:

$$\Phi(f_1,\ldots,f_m) - \Phi(f_1,f_2,\ldots,f_{j-1},f_j',f_{j+1},\ldots,f_m)$$

$$= \sum_{i''=1}^{K} p_{i''} \cdot G_{i''}(f_1,\ldots,f_m) - \sum_{i''=1}^{K} p_{i''} \cdot G_{i''}(f_1,f_2,\ldots,f_{j-1},f_j',f_{j+1},\ldots,f_m)$$

$$= p_i \cdot \big(G_i(f_1,\ldots,f_m) - G_i(f_1,f_2,\ldots,f_{j-1},f_j',f_{j+1},\ldots,f_m)\big)$$

$$+ p_{i'} \big(G_{i'}(f_1,\ldots,f_m) - G_{i'}(f_1,f_2,\ldots,f_{j-1},f_j',f_{j+1},\ldots,f_m)\big).$$

If $N_i > 1$, then:

$$G_i(f_1,\ldots,f_m) - G_i(f_1,f_2,\ldots,f_{j-1},f_j',f_{j+1},\ldots,f_m) = \frac{1}{N^i}$$

and if $N_i = 1$, then

$$G_i(f_1,\ldots,f_m) - G_i(f_1,f_2,\ldots,f_{j-1},f_j',f_{j+1},\ldots,f_m) = 1 - \frac{1}{m}.$$

Similarly, if $N_{i'} > 0$, then:

$$G_{i'}(f_1,\ldots,f_m) - G_{i'}(f_1,f_2,\ldots,f_{j-1},f_j',f_{j+1},\ldots,f_m) = -\frac{1}{N^{i'}+1}$$

and if $N_{i'} = 0$, then

$$G_{i'}(f_1,\ldots,f_m) - G_i(f_1,f_2,\ldots,f_{j-1},f_j',f_{j+1},\ldots,f_m) = -\left(1 - \frac{1}{m}\right).$$

Altogether, this implies that:

$$\Phi(f_1,\ldots,f_m) - \Phi(f_1,f_2,\ldots,f_{j-1},f_j',f_{j+1},\ldots,f_m) = u(f_j;\mathbf{f}_{-j}) - u(f_j';\mathbf{f}_{-j}),$$

which shows that $\Phi$ is a potential function of the game. Since pure strategy equilibria exist in potential games [Rosenthal, 1973, Monderer and Shapley, 1996], a pure strategy equilibrium must exist in the game. $\qquad\square$

## D.2 Proofs for Section 3.1

We next prove Proposition 2. The high-level intuition of the proof is as follows. By Lemma 5, we can focus on one data representation $x$ at a time. Let $y^* = \arg\max_y \mathbb{P}[y \mid x]$ be the Bayes optimal label of $x$. The proof boils down to characterizing when the market outcome, $f_j(x) = y^*$ for $j \in [m]$, is an equilibrium, and the equilibrium social loss is determined by whether this market outcome is an equilibrium or not.

*Proof of Proposition 2.* Let $\mathcal{D}^R$ be the marginal distribution of $X$ with respect to the distribution $(X,Y) \sim \mathcal{D}$. Let $f_1,\ldots,f_m$ be a pure-strategy equilibrium. The social loss is equal to:

$$\mathtt{SL}(f_1,\ldots,f_m) = \mathbb{E}[\ell(f_{j^*(x,y)}(x),y)]$$

$$= \mathop{\mathbb{E}}_{x'\sim\mathcal{D}^R}\left[\mathop{\mathbb{E}}_{(x,y)\sim\mathcal{D}}[\ell(f_{j^*(x,y)}(x),y) \mid x = x']\right]$$

$$= \mathop{\mathbb{E}}_{x'\sim\mathcal{D}^R}\left[\mathop{\mathbb{E}}_{(x,y)\sim\mathcal{D}_{x'}}[\ell(f_{j^*(x,y)}(x),y)]\right],$$

where $\mathcal{D}_{x'}$ denotes the conditional distribution $(X,Y) \mid X = x'$ where $(X,Y) \sim \mathcal{D}$. Thus, to analyze the overall social loss, we can separately analyze the social loss on each distribution $\mathcal{D}_{x'}$ and then average across distributions. It suffices to show that $\mathbb{E}_{\mathcal{D}_{x'}}[\ell(f_{j^*(x,y)}(x),y)] = \alpha(x')$ if $\alpha(x') < 1/m$ and zero if $\alpha(x') > 1/m$.

To compute the social loss on $\mathcal{D}_{x'}$, we first apply Lemma 5. This means that $(f_1(x'),\ldots,f_m(x'))$ is pure-strategy equilibrium with $\mathcal{D}_{x'}$. We characterize the equilibrium structure for $\mathcal{D}_{x'}$ and use this characterization to compute the equilibrium social loss.

**Equilibrium structure for $\mathcal{D}_{x'}$.** For notational convenience, let $y_i := f_i(x')$ denote the label chosen by model-provider $i$ and let let $y^* = \arg\max_y \mathbb{P}[y \mid x']$ be the Bayes optimal label for $x'$. We also abuse notation slightly and let $u(y_1; y_{-j})$ be model-provider 1's utility if they choose the label $y_1$ for $x'$ and the other model-provider's choose $y_{-j}$.

We first show that all model-providers choosing $y^*$ is an equilibrium if and only if $\alpha(x') \leq 1/m$. Let's fix $y_j = y^*$ for all $j \geq 2$ and look at model-provider 1's utility. We see that $u(y^*; y_{-j}) = 1/m$ and $u(1 - y^*; y_{-j}) = \alpha(x')$. This means that $y^*$ is a best-response (i.e., $y^* \in \arg\max_y u(y; y_{-j})$) if and only if $\alpha(x') \leq 1/m$.

We next show that if $\alpha(x') < 1/m$, then the market outcome $y_i = y^*$ for all $i \in [m]$ is the only pure-strategy equilibrium. Let $y_1, \ldots, y_m$ be a pure-strategy equilibrium. It suffices to show that $y^*$ is the unique best response to $y_{-j}$; that is, that $\{y^*\} = \arg\max_y u(y; y_{-j})$. To show this, let $m'$ denote the size of the set $\{2 \leq i \leq m \mid y_i = y^*\}$. First, if $m' = 0$, then we have that

$$u(y^*; y_{-j}) = 1 - \alpha(x') > 1/m = u(1 - y^*; y_{-j}),$$

where $1 - \alpha(x') > 1/m$ follows from the fact that $1 - \alpha(x') \geq 1/2 \geq 1/m$ along with our assumption that $\alpha(x') \neq 1/m$. This demonstrates that $y^*$ is indeed the unique best response. If $m' = m - 1$, then we have that:

$$u(y^*; y_{-j}) = 1/m > \alpha(x') = u(1 - y^*; y_{-j}),$$

as desired. Finally, if $1 \leq m' \leq m - 2$, then:

$$u(y^*; y_{-j}) = \frac{1 - \alpha(x')}{m' + 1} \geq \frac{1 - \alpha(x')}{m - 1} > \frac{1}{m} > \alpha(x') > \frac{\alpha(x')}{m - m'} = u(1 - y^*; y_{-j}),$$

as desired.

Finally, we show that all model-providers choosing $1 - y^*$ is never an equilibrium. Let's fix $y_j = 1 - y^*$ and look at model-provider 1's utility. We see that:

$$u(y^*; y_{-j}) = 1 - \alpha(x') > \frac{\alpha(x')}{m} = u(1 - y^*; y_{-j}),$$

which shows that $y^*$ is the unique best response as desired.

**Characterization of equilibrium social loss.** It follows from (4) that the equilibrium social loss $\mathbb{E}_{(x,y) \sim \mathcal{D}_{x'}}[\ell(f_{j^*(x,y)}(x), y)]$ is $\alpha(x')$ if all of the model-providers choose $y_i = y^*$, it is zero if a nonzero number of model-providers choose $y^*$ and a nonzero number of model-providers choose $1 - y^*$, and it is $1 - \alpha(x')$ if all of the model-providers choose $1 - y^*$.

Let's combine this with our equilibrium characterization results. If $\alpha(x') < 1/m$, then the unique equilibrium is at $y_i = y^*$ so the equilibrium social loss is $\alpha(x)$ as desired. If $\alpha(x') > 1/m$, then neither $y_i = y^*$ for all $i \in [m]$ nor $y_i = 1 - y^*$ for all $i \in [m]$ is an equilibrium. Since there exists a pure strategy equilibrium by Proposition 1, there must be a pure strategy equilibrium where a nonzero number of model-providers choose $y^*$ and a nonzero number of model-providers choose $1 - y^*$. The equilibrium social loss is thus zero.

Note when $\alpha(x') = 1 - 1/m$, there is actually an equilibrium where all of the model-providers choose $y_i = y^*$, 0 and an equilibrium where a nonzero number of model-providers choose $y^*$ and a nonzero number of model-providers choose $1 - y^*$; thus, the equilibrium social loss can be zero or $1/m$.

□

## D.3 Proofs for Section 3.3

We prove Proposition 3.

*Proof of Proposition 3.* Let $\mathcal{D}^R$ be the marginal distribution of $X$ with respect to the distribution $(X, Y) \sim \mathcal{D}$. Let $f_1, \ldots, f_m$ be a pure-strategy equilibrium. The social loss is equal to:

$$\mathrm{SL}(f_1, \ldots, f_m) = \mathbb{E}[\ell(f_{j^*(x,y)}(x), y)]$$

$$= \mathbb{E}_{x' \sim \mathcal{D}^R} \left[ \mathbb{E}_{(x,y) \sim \mathcal{D}} [\ell(f_{j^*(x,y)}(x), y) \mid x = x'] \right]$$

$$= \mathbb{E}_{x' \sim \mathcal{D}^R} \left[ \mathbb{E}_{(x,y) \sim \mathcal{D}_{x'}} [\ell(f_{j^*(x,y)}(x), y)] \right],$$

where $\mathcal{D}_{x'}$ denotes the conditional distribution $(X, Y) \mid X = x'$ where $(X, Y) \sim \mathcal{D}$. Thus, to analyze the overall social loss, we can separately analyze the social loss on each distribution $\mathcal{D}_{x'}$ and then average across distributions. It suffices to show that

$$\mathbb{E}_{\mathcal{D}_{x'}} \left[ \sum_{i=1}^{K} \alpha^i(x) \cdot \mathbb{1} \left[ \alpha^i(x) < \frac{c}{m} \right] \right] \leq \mathbb{E}_{\mathcal{D}_{x'}} [\ell(f_{j^*(x,y)}(x), y)] \leq \mathbb{E}_{\mathcal{D}_{x'}} \left[ \sum_{i=1}^{K} \alpha^i(x) \cdot \mathbb{1} \left[ \alpha^i(x) \leq \frac{1}{m} \right] \right].$$

To compute the social loss on $\mathcal{D}_{x'}$, we first apply Lemma 5. This means that $(f_1(x'), \ldots, f_m(x'))$ is pure-strategy equilibrium with $\mathcal{D}_{x'}$. We then prove properties of the equilibrium structure for $\mathcal{D}_{x'}$ and use these properties to bound the equilibrium social loss. For notational convenience, let $y_i := f_i(x')$ denote the label chosen by model-provider $i$ and let let $y^* = \arg\max_y \mathbb{P}[y \mid x']$ be the Bayes optimal label for $x'$. We also abuse notation slightly and let $u(y_1; y_{-j})$ be model-provider 1's utility if they choose the label $y_1$ for $x'$ and the other model-provider's choose $y_{-j}$. We can rewrite:

$$\mathbb{E}_{\mathcal{D}_{x'}} [\ell(f_{j^*(x,y)}(x), y)] = \mathbb{E}_{\mathcal{D}_{x'}} \left[ \sum_{i=1}^{K} \alpha^i(x) \cdot \mathbb{1} [y_j \neq i \text{ for all } j \in [m]] \right].$$

We first prove the lower bound on $\mathbb{E}_{\mathcal{D}_{x'}} [\ell(f_{j^*(x,y)}(x), y)]$ and then we prove the upper bound on $\mathbb{E}_{\mathcal{D}_{x'}} [\ell(f_{j^*(x,y)}(x), y)]$.

**Proof of lower bound.** Let $y_1, \ldots, y_m$ be a pure strategy equilibrium. To prove the lower bound, it suffices to show that if $\alpha^i(x) < c/m$, then $y_j \neq i$ for all $j \in [m]$.

Assume for sake of contradiction that $\alpha^i(x) < c/m$ and $y_j = i$ for some $j \in [m]$. Let $i' = \arg\max_{i'' \in \{0, 1, \ldots, K-1\}} \alpha^{i''}(x)$ be the class with maximal conditional probability. By the definition of $c$, we see that $\alpha^{i'}(x) \geq c > c/m$ which also implies that $i' \neq i$. We split into two cases—(1) $y_{j'} \neq i'$ for all $j' \in \{0, 1, \ldots, K-1\}$, and (2) $y_{j'} = i'$ for some $j' \in \{0, 1, \ldots, K-1\}$—and derive a contradiction in each case.

Consider the first case where $y_{j'} \neq i'$ for all $j' \in \{0, 1, \ldots, K-1\}$. Then if model-provider $j$ switched from $y_j$ to $i'$, the difference in their utility would be bounded as:

$$u(i'; y_{-j}) - u(y_j; y_{-j}) \geq \alpha^{i'}(x) - \left( \frac{\alpha^{i'}(x)}{m} + \alpha^i(x) \right)$$

$$= \alpha^{i'}(x) \left( 1 - \frac{1}{m} \right) - \alpha^i(x)$$

$$> c \left( 1 - \frac{1}{m} \right) - \frac{c}{m}$$

$$= c \left( 1 - \frac{2}{m} \right)$$

$$\geq 0,$$

so $y_j$ is not a best-response for model-provider $j$, which is a contradiction.

Now, consider the second case, where $y_{j'} = i'$ for some $j' \in \{0, 1, \ldots, K-1\}$. If we compare the utility when model-provider $j$ chooses $i'$ versus $y_j$ as their action, the difference is utility can be bounded as:

$$u(i'; y_{-j}) - u(y_j; y_{-j}) \geq \frac{\alpha^{i'}(x)}{m} - \alpha^i(x) > \frac{c}{m} - \frac{c}{m} = 0.$$

|  | $y_2 = 1 - y^*$ | $y_2 = y^*$ |
|---|---|---|
| $y_1 = 1 - y^*$ | $(w_{\max}, w_{\min})$ | $(\alpha(x), 1 - \alpha(x))$ |
| $y_1 = y^*$ | $(1 - \alpha(x), \alpha(x))$ | $(w_{\max}, w_{\min})$ |

Table 1: Let $X = \{x\}$, $\mathcal{F} = \mathcal{F}_{\text{all}}^{\text{binary}}$, user decisions are noiseless, and user decisions are noiseless (i.e., $c \to 0$, so user decisions are given by (7)). Suppose that there are $m = 2$ model-providers with market reputations $w_{\min}$ and $w_{\max}$, where $w_{\max} \geq w_{\min}$ and $w_{\max} + w_{\min} = 1$. Let $y^* = \arg\max_y \mathbb{P}[y \mid x]$ be the Bayes optimal label for $x'$. The table shows the game matrix when model-provider 1 chooses the label $y_1$ and model provider 2 chooses the label $y_2$.

so $y_j$ is not a best-response for model-provider $j$, which is a contradiction.

This proves the lower bound as desired.

**Proof of upper bound.** Let $y_1, \ldots, y_m$ be a pure strategy equilibrium. To prove the upper bound, it suffices to show if $\alpha^i(x) > 1/m$, then $y_j = i$ for some $j \in [m]$. Assume for sake of contradiction that $\alpha^i(x) > 1/m$ and $y_j \neq i$ for all $j \in [m]$. For any set of actions $y_1, \ldots, y_m$, the total utility $\sum_{j=1}^m u(y_j; y_{-j}) = 1$ sums to 1. Thus, some model provider $j \in [m]$ must have utility satisfying $u(y_j; y_{-j}) \leq 1/m$. However, if model-provider $j$ instead chose action $i$, then they would achieve utility:

$$u(i; y_{-j}) \geq \alpha^i(x) > \frac{1}{m} \geq u(y_j; y_{-j}),$$

so $y_j$ is not a best-response for model-provider $j$, which is a contradiction. This proves the upper bound as desired.

$\square$

### D.4 Proofs for Section B.1

A useful lemma is the following calculation of the game matrix when there is a single representation $X = \{x\}$.

**Lemma 6.** *Let $X = \{x\}$, and let $\mathcal{F} = \mathcal{F}_{\text{all}}^{\text{binary}}$. Suppose that there are $m = 2$ model-providers with market reputations $w_{\min}$ and $w_{\max}$, where $w_{\max} \geq w_{\min}$ and $w_{\max} + w_{\min} = 1$. Suppose that user decisions are noiseless (i.e., $c \to 0$, so user decisions are given by (7)). Then the game matrix is specified by Table 1.*

*Proof.* This follows from applying (7) and using the fact that $\ell(y, y') = \mathbb{1}[y \neq y']$. $\square$

We show that pure strategy equilibria are no longer guaranteed to exist when model-providers have unequal market reputations, even when there is a single representation $X = \{x\}$.

**Lemma 7.** *Let $X = \{x\}$ let $\mathcal{F} = \mathcal{F}_{\text{all}}^{\text{binary}}$. Suppose that there are $m = 2$ model-providers with market reputations $w_{\min}$ and $w_{\max}$, where $w_{\max} \geq w_{\min}$ and $w_{\max} + w_{\min} = 1$. Suppose that user decisions are noiseless (i.e., $c \to 0$, so user decisions are given by (7)). If $\alpha(x) > w_{\min}$, then a pure strategy equilibrium does not exist.*

*Proof.* For notational convenience, let $y_i := f_i(x')$ denote the label chosen by model-provider $i$ and let $y^* = \arg\max_y \mathbb{P}[y \mid x']$ be the Bayes optimal label for $x'$. We also abuse notation slightly and let $u_i(y; y')$ be model-provider $i$'s utility if they choose the label $y$ for $x$ and the other model-providers choose $y'$. The proof follows from the game matrix show in Table 1 (Lemma 6). Using the fact that model-provider 1 must best-respond to model-provider 2's action, this leaves $y_1 = 1 - y^*$, $y_2 = 1 - y^*$ and $y_1 = y^*$, $y_2 = y^*$. However, neither of these market outcomes captures a best-response for model-provider 2: if $y_1 = 1 - y^*$, then model-provider 2's unique best response is $y^*$; if $y_1 = y^*$, then model-provider 2's unique best response is $1 - y^*$. This rules out the existence of a symmetric or asymmetric pure strategy equilibrium. $\square$

Given the lack of existence of pure strategy equilibria, we must turn to mixed strategies. A mixed strategy equilibrium is guaranteed to exist since the game has finitely many actions $\mathcal{F}_{\text{all}}^{\text{binary}}$ and finitely many players $m$. Let $(\mu_1, \mu_2, \ldots, \mu_m)$ denote a mixed strategy profile over $\mathcal{F}_{\text{all}}^{\text{binary}}$. We show the following analogue of Lemma 5 that allows us to again decompose model-provider actions into independent decisions about each representation $x$. To formalize this, let $\mathcal{D}$ be the data distribution, and again let $\mathcal{D}_x$ be the conditional distribution of $(X, Y)$ when $X = x$, where $(X, Y) \sim \mathcal{D}$. Again, let $(\mathcal{F}_{\text{all}}^{\text{binary}})^x := \{f_0, f_1\}$ be the class of the (two) functions from a single representation $x$ to $\{0, 1\}$, where $f_0(x) = 0$ and $f_1(x) = 1$. Given a mixed strategy profile $\mu$ and a representation $x$, we define the conditional mixed strategy $\mu^x$ over $(\mathcal{F}_{\text{all}}^{\text{binary}})^x := \{f_0, f_1\}$ to be defined so $\mathbb{P}_{\mu^x}[f_i] := \mathbb{P}_{f \sim \mu}[f(x) = i]$ for $i \in \{0, 1\}$.

**Lemma 8.** *Let $X$ be a finite set of representations, let $\mathcal{F} = (\mathcal{F}_{\text{all}}^{\text{binary}})$, and let $\mathcal{D}$ be the distribution over $(X, Y)$. For each $x \in X$, let $\mathcal{D}_x$ be the conditional distribution of $(X, Y)$ given $X = x$, where $(X, Y) \sim \mathcal{D}$, and let $(\mathcal{F}_{\text{all}}^{\text{binary}})^x := \{f_0, f_1\}$ be the class of the (two) functions from a single representation $x$ to $\{0, 1\}$, where $f_0(x) = 0$ and $f_1(x) = 1$. Suppose that user decisions are noiseless (i.e., $c \to 0$, so user decisions are given by (3)). A strategy profile $(\mu_1, \mu_2, \ldots, \mu_m)$ is an equilibrium if and only if for every $x \in X$, the market outcome $(\mu_1^x, \mu_2^x, \ldots, \mu_m^x)$ (where $\mu_1^x, \ldots, \mu_m^x$ are the conditional mixed strategies defined above) is an equilibrium for $(\mathcal{F}_{\text{all}}^{\text{binary}})^x$ with data distribution $\mathcal{D}_x$.*

*Proof.* The proof follows similarly to the proof of Lemma 8, but some minor generalizations to account for mixed strategy equilibria. Let $\mathcal{D}^R$ be the marginal distribution of $X$ with respect to the distribution $(X, Y) \sim \mathcal{D}$. Let $\mathcal{D}^R$ be the marginal distribution of $X$ with respect to the distribution $(X, Y) \sim \mathcal{D}$. First, we write model-provider $j$'s utility as:

$$
\mathbb{E}_{\substack{f_j \sim \mu_j \\ \mathbf{f}_{-j} \sim \mu_{-j}}} [u(f_j; \mathbf{f}_{-j})] = \mathbb{E}_{\substack{f_j \sim \mu_j \\ \mathbf{f}_{-j} \sim \mu_{-j}}} \left[ \mathbb{E}_{(x,y) \sim \mathcal{D}} [\mathbb{P}[j^*(x, y) = j]] \right] = \mathbb{E}_{x' \sim \mathcal{D}^R} \left[ \mathbb{E}_{\substack{f_j \sim \mu_j^{x'} \\ \mathbf{f}_{-j} \sim \mu_{-j}^{x'}}} \left[ \mathbb{E}_{(x,y) \sim \mathcal{D}_{x'}} [\mathbb{P}[j^*(x, y) = j]] \right] \right].
$$

$$(10)$$

where $\mu_{-j}$ denotes the mixed strategies chosen by the other model-providers.

First we show that if $\mu_1, \mu_2, \ldots, \mu_m$ is an equilibrium, then $(\mu_1^{x'}, \ldots, \mu_m^{x'})$ is an equilibrium for $(\mathcal{F}_{\text{all}}^{\text{binary}})^{x'}$ with data distribution $\mathcal{D}_{x'}$. Let $f_j$ be in supp$(\mu_{j'})$. Assume for sake of contradiction that $(\mu_1^{x'}, \ldots, \mu_m^{x'})$ is not an equilibrium. Then there exists $j' \in [m]$ such that model-provider $j'$ would achieve higher utility on $f^{1-f_{j'}(x')}$ than $f^{f_{j'}(x')}$. Let $f'_{j'}$ be the predictor given by $f'_{j'}(x) = f_{j'}(x)$ if $x \neq x'$ and $f'_{j'}(x') = 1 - f_{j'}(x')$. By equation (10), this would mean that $u(f'_{j'}; \mu_{-j'})$ is strictly higher than $u(f_{j'}; \mu_{-j'})$ which is a contradiction.

Next, we show that if $(\mu_1^{x'}, \ldots, \mu_m^{x'})$ is an equilibrium for $(\mathcal{F}_{\text{all}}^{\text{binary}})^{x'}$ with data distribution $\mathcal{D}_{x'}$ for all $x' \in X$ then $\mu_1, \ldots, \mu_m$ is an equilibrium. Let $f_j$ be in supp$(\mu_{j'})$. Assume for sake of contradiction that there exists $j'$ such that $u(f'_{j'}; \mu_{-j'}) > u(f_j; \mu_{-j'})$. By equation (9), there must exist $x'$ such that $\mathbb{E}_{\mathbf{f}_{-j'} \sim \mu_{-j'}^{x'}} \left[ \mathbb{E}_{(x,y) \sim \mathcal{D}_{x'}} [\mathbb{P}[j^*(x, y) = j']] \right]$ is higher for $f'_{j'}$ than $f_{j'}$. This means that $(\mu_1^{x'}, \ldots, \mu_m^{x'})$ is not an equilibrium, which is a contradiction. $\square$

We now prove Proposition 4.

*Proof of Proposition 4.* Let $\mathcal{D}^R$ be the marginal distribution of $x$ with respect to the distribution $(x, y) \sim \mathcal{D}$. Let $\mu_1, \mu_2$ be a mixed strategy equilibrium. The social loss is equal to:

$$\underset{\substack{f_1 \sim \mu_1 \\ f_2 \sim \mu_2}}{\mathbb{E}} [\mathrm{SL}(f_1, f_2)] = \mathbb{E}[\ell(f_{j^*(x,y)}(x), y)]$$

$$= \underset{\substack{f_1 \sim \mu_1 \\ f_2 \sim \mu_2}}{\mathbb{E}} \left[ \underset{x' \sim \mathcal{D}^R}{\mathbb{E}} \left[ \underset{(x,y) \sim \mathcal{D}}{\mathbb{E}} [\ell(f_{j^*(x,y)}(x), y) \mid x = x'] \right] \right]$$

$$= \underset{\substack{f_1 \sim \mu_1 \\ f_2 \sim \mu_2}}{\mathbb{E}} \left[ \underset{x' \sim \mathcal{D}^R}{\mathbb{E}} \left[ \underset{(x,y) \sim \mathcal{D}_{x'}}{\mathbb{E}} [\ell(f_{j^*(x,y)}(x), y)] \right] \right]$$

$$= \underset{x' \sim \mathcal{D}^R}{\mathbb{E}} \left[ \underset{\substack{f_1 \sim \mu_1^* \\ f_2 \sim \mu_2^*}}{\mathbb{E}} \left[ \underset{(x,y) \sim \mathcal{D}_{x'}}{\mathbb{E}} [\ell(f_{j^*(x,y)}(x), y)] \right] \right]$$

$$= \underset{x' \sim \mathcal{D}_X}{\mathbb{E}} \left[ \underset{\substack{f_1 \sim \mu_1^{x'} \\ f_2 \sim \mu_2^{x'}}}{\mathbb{E}} \left[ \underset{(x,y) \sim \mathcal{D}_{x'}}{\mathbb{E}} [\ell(f_{j^*(x,y)}(x), y)] \right] \right]$$

where $\mathcal{D}_{x'}$ denotes the conditional distribution $(X, Y) \mid X = x'$ where $(X, Y) \sim \mathcal{D}$ and where $\mu^x$ denotes the conditional mixed strategy $(\mathcal{F}_{\mathrm{all}}^{\mathrm{binary}})^x := \{f^0, f^1\}$ to be defined so $\mathbb{P}_{\mu^x}[f^i] := \mathbb{P}_{f \sim \mu}[f(x) = i]$ for $i \in \{0, 1\}$ Thus, to analyze the overall social loss, we can separately analyze the social loss on each distribution $\mathcal{D}_{x'}$ and then average across distributions. It suffices to show that:

$$\underset{\substack{f_1 \sim \mu_1^{x'} \\ f_2 \sim \mu_2^{x'}}}{\mathbb{E}} \left[ \underset{(x,y) \sim \mathcal{D}_{x'}}{\mathbb{E}} [\ell(f_{j^*(x,y)}(x), y)] \right] = \begin{cases} \alpha(x') & \text{if } \alpha(x') < w_{\min} \\ \frac{2(\alpha(x') - w_{\min}) \cdot (w_{\max} - \alpha(x))}{(1 - 2 \cdot w_{\min})^2} & \text{if } \alpha(x') > w_{\min}. \end{cases}$$

To compute the social loss on $\mathcal{D}_{x'}$, we first apply Lemma 8. This means that $(\mu_1^{x'}, \mu_2^{x'})$ is mixed-strategy equilibrium with $\mathcal{D}_{x'}$. We characterize the equilibrium structure for $\mathcal{D}_{x'}$ and use this characterization to compute the equilibrium social loss.

Our main technical ingredient is the game matrix in Table 1 (Lemma 6). We will slightly abuse notation and view choosing the label $y$ as the strategy of the model-provider. Accordingly, we view a mixed strategy as a distribution over $\{0, 1\}$. For notational convenience, let $y_i := f_i(x')$ denote the label chosen by model-provider $i$ and let $y^* = \arg\max_y \mathbb{P}[y \mid x']$ be the Bayes optimal label for $x'$. We split into two cases: $\alpha(x') < w_{\min}$ and $\alpha(x') > w_{\min}$.

**Case 1:** $\alpha(x') < w_{\min}$. We claim that the unique equilibrium is a pure strategy equilibrium where $y_1 = y_2 = y^*$. First, if $\alpha(x) < w_{\min}$, we show that choosing $y^*$ is a strictly dominant strategy for model-provider 1. This follows from the fact that $1 - \alpha(x) > w_{\max}$ and $w_{\max} \geq w_{\min} > \alpha(x)$. Thus, model-provider 1 must play a pure strategy where they always choose $y_1 = y^*$. When model-provider 1 chooses $y^*$, then the unique best response for model-provider 2 is also to choose $y^*$ since $\alpha(x') < w_{\min}$. This establishes that $y_1 = y_2 = y^*$ is the unique equilibrium. This also implies that the equilibrium social loss satisfies:

$$\underset{\substack{f_1 \sim \mu_1^{x'} \\ f_2 \sim \mu_2^{x'}}}{\mathbb{E}} \left[ \underset{(x,y) \sim \mathcal{D}_{x'}}{\mathbb{E}} [\ell(f_{j^*(x,y)}(x), y)] \right] = \alpha(x')$$

as desired.

**Case 2:** $\alpha(x') > w_{\min}$. Let $p_1 = \mathbb{P}_{\mu_1^{x'}}[y_1 = y^*]$ and let $p_2 = \mathbb{P}_{\mu_2^{x'}}[y_2 = y^*]$. By Lemma 7, a pure strategy equilibrium does not exist. Thus, we consider mixed strategies. Since pure strategy equilibria do not exist, at least one of $p_1$ and $p_2$ must be strictly between zero and one. We compute $p_1$ and $p_2$, splitting into two cases: (1) $p_1 > 0$ and (2) $p_2 > 0$.

If $p_1 > 0$, then we know that model-provider 1 must be indifferent between choosing $y^*$ and $1 - y^*$. This means that:

$$p_2\alpha(x') + (1 - p_2)w_{\max} = (1 - p_2)(1 - \alpha(x')) + p_2 w_{\max}.$$

Solving for $p_2$, we obtain:

$$p_2 = \frac{w_{\max} - (1 - \alpha(x'))}{2w_{\max} - 1} = \frac{\alpha(x') - w_{\min}}{1 - 2w_{\min}} > 0.$$

If $p_2 > 0$, then we know that model-provider 2 must be indifferent between choosing $y^*$ and $1 - y^*$. This means that:

$$p_1\alpha(x') + (1 - p_1)w_{\min} = (1 - p_1)(1 - \alpha(x')) + p_1 w_{\min}.$$

Solving for $p_1$, we obtain:

$$p_1 = \frac{(1 - \alpha(x')) - w_{\min}}{1 - 2w_{\min}} = \frac{w_{\max} - \alpha(x)}{1 - 2w_{\min}} > 0.$$

Putting this all together, we see that:

$$p_1 = \frac{w_{\max} - \alpha(x')}{1 - 2w_{\min}}$$
$$p_2 = \frac{\alpha(x') - w_{\min}}{1 - 2w_{\min}},$$

and in fact $p_1 + p_2 = 1$.

Using this characterization of $p_1$ and $p_2$, we see that the equilibrium social loss is equal to:

$$\mathop{\mathbb{E}}_{\substack{f_1 \sim \mu_1^{x'} \\ f_2 \sim \mu_2^{x'}}}\left[\mathop{\mathbb{E}}_{(x,y) \sim \mathcal{D}_{x'}}[\ell(f_{j^*(x,y)}(x), y)]\right] = \alpha(x')\mathbb{P}[y_1 = y^*]\mathbb{P}[y_2 = y^*] + (1 - \alpha(x'))\mathbb{P}[y_1 = 1 - y^*]\mathbb{P}[y_2 = 1 - y^*]$$

$$= \alpha(x')p_1 p_2 + (1 - \alpha(x))(1 - p_1)(1 - p_2)$$
$$= \alpha(x')p_1 p_2 + (1 - \alpha(x))p_1 p_2$$
$$= p_1 p_2$$
$$= \frac{(\alpha(x') - w_{\min}) \cdot (w_{\max} - \alpha(x))}{(1 - 2 \cdot w_{\min})^2},$$

as desired.

□