# OpenReview forum: "Improved Bayes Risk Can Yield Reduced Social Welfare Under Competition"
_NeurIPS.cc/2023/Conference — NeurIPS 2023 poster_

### Official Review · Reviewer_tyS5 · 2023-07-06

**Soundness:** 3 good
**Presentation:** 3 good
**Contribution:** 2 fair
**Rating:** 6
**Confidence:** 4

**Summary:**

This study highlights the inconsistencies of scaling laws in machine learning when model providers compete with each other in the market. The researchers conducted experiments with varying data representations and found that improving data representation quality may actually decrease the overall predictive accuracy, which is modeled as social welfare, across users in the competitive market. They also generalized experiment to unfair markets and study the market with 2 model providers only. The study raises awareness about the importance of evaluating machine learning models in a market where organizations are competing.

**Strengths:**

This paper presents several strengths, including a clear and well-written introduction that guides the reader to understand their interesting findings and the awareness behind it. Approaches to vary data representation are comprehensive and practical. They provide basic setups to study and offer results empirically and theoretically. Notably, the paper's conclusion offers an interesting insight into why enhancing model accuracy may not necessarily lead to significant social benefits in real-world applications.

**Weaknesses:**

Here are some potential improvements to the original content:

1. The available training resources of model providers in the market are unclear, as well as the basis on which they build their models. If models are trained solely on user-provided datasets, their performance may not vary significantly, which may make the best response trivial.

2. The experiment only covers CIFAR-10 and reduces it to a binary classification task, which warrants further exploration with datasets of higher cardinality.

3. The finding about inefficient translation to social welfare is interesting, but it lacks proper and thorough explanation.

**Questions:**

If the market is competitive, when and why do people care about the social welfare? Organizations are selfish as they only care about earning more share in the market as much as possible. Is there any scenario in practice in machine learning that people care about social welfare more than their own preference or happiness.

**Limitations:**

This paper presents a novel finding that challenges the conventional understanding of scaling laws in machine learning in the context of organizational competition. The framework proposed is foundational and can be applied to various scenarios.

---

> ### Author Rebuttal · Authors · 2023-08-10
>
> We thank the reviewer for their feedback.  We present **new results** that address some concerns and also respond to other concerns.
>
> **“The experiment only covers CIFAR-10 and reduces it to a binary classification task, which warrants further exploration with datasets of higher cardinality.”**
>
> We provide new results showing that our empirical results in Section 4.3 directly extend to CIFAR-10 with 10 classes, and our theoretical results in Section 3.1 directly extend to multi-class classification with K classes. (Our motivation for studying binary classification was to simplify the exposition and analysis.)
>
> In **new multi-class experiments on CIFAR-10**, we consider a data distribution over 50,000 samples and the original 10 classes. We show that the equilibrium social loss is non-monotonic in the data representation quality as measured by Bayes risk (see Figure N1 in the General Response pdf). Figure N1 shows that the empirical results in Section 4.3 directly extend to CIFAR-10 with 10 classes.
>
> In a new theoretical result, we show that Proposition 1 can be directly extended to multi-class classification, where the function class is all functions $f:X \rightarrow Y$ where $Y = [K]$. The same insights from Section 3.1 thus carry over to multi-class classification.
>
> For details of the new experiments and new theoretical result, we refer the reviewer to the "Extension to classification with more than 2 classes" section of the General Response.
>
> **“If the market is competitive, when and why do people care about social welfare? Is there any scenario in practice in machine learning that people care about social welfare more than their own preference or happiness?”**
>
> The reviewer is correct that every *individual user* cares only about their own preferences and happiness, which is why the user chooses the model-provider that maximizes that user’s happiness (see discussion below for more details). However, when model-providers compete to attract the most users (line 124), our results show that *society* (i.e., the user population as a whole) becomes worse off. Our measure of social welfare exactly captures the *overall happiness of the user population as a whole*.
>
> This measure of social welfare is the typical metric studied in the existing work on competing predictors (e.g. Ben-Porat and Tenneholtz [2017, 2019]) and is ultimately what is important from a societal perspective (e.g., Kleinberg and Raghavan 2021). We envision that policymakers and regulators would care about social welfare of the user population and may develop interventions to ensure that social welfare is high.
>
> In more detail, to see why our social welfare metric exactly captures the overall happiness of the user population, we refer the reviewer to Section 2 and summarize the main points below.
> - The  user utility function implicit in this definition is $-\ell(y, f(x))$ where $y$ is the user’s true label and $f(x)$ is the prediction that they receive. In other words, user utility is a negative of the loss that they receive for their prediction.
> - Each user chooses the model-provider $j^*(x,y)$ (see line 120) with predictor $f_{j^*(x,y)}$ that maximizes that user’s utility.
> - Recall that we measure social welfare by the “overall predictive accuracy” for users (see definition of social loss in line 141). Social welfare thus measures the overall happiness (total user utility) of all users in the population.
>
> **“If models are trained solely on user-provided datasets, their performance may not vary significantly, which may make the best response trivial.”**
>
>  This intuition actually turns out to be incorrect: even if the models are trained on the same data distribution and dataset, the model-providers may select different predictors. The reason is that optimizing for market share is  fundamentally different from optimizing for empirical risk. As a result, model-providers may not choose the same predictor, or even choose Bayes optimal predictors, at equilibrium ((Proposition 1, Section 3.2, Section 4, Figure 1).
>
> **“The finding about inefficient translation to social welfare is interesting, but it lacks proper and thorough explanation.”**
>
> We are unsure what form of explanation the reviewer has in mind. The paper includes formal details of the model and assumptions (Section 2), theoretical justification of our results (Section 3), and empirical evaluation of our results (Section 4). Furthermore, we have included additional experiments for multi-class classification (see above and the General Response) and local improvements to representations (see the General Response). We would be happy to provide further explanations during the discussion period and to the final version of the paper if the reviewer could provide more details about what they find lacking.

---

> > ### Comment · Reviewer_tyS5 · 2023-08-15
> > **Thanks for your detailed clarification!!**
> >
> > Dear Authors,
> >
> > Thanks for your clarifications and additional experimental results. The new cifar10 results look good and comprehensive. Your further explanations on social welfare have addressed my concerns. About the "inefficient translation to social welfare", I have read this again in the paper and corrected my understanding. Thanks for response overall. I have raised my score by 1 and support this paper to be accepted.

---

### Official Review · Reviewer_YGiJ · 2023-07-10

**Soundness:** 3 good
**Presentation:** 2 fair
**Contribution:** 3 good
**Rating:** 6
**Confidence:** 2

**Summary:**

The paper explores the impact of competition on machine learning models and social welfare. It investigates how scaling trends can be altered by competition and how improving data representation quality may not always lead to better predictive accuracy. The authors provide a theoretical framework for analyzing the impact of competition on machine learning models and demonstrate their findings through experiments on real-world datasets. The paper makes contributions to the understanding of the relationship between competition, data representation quality, and predictive accuracy in machine learning models.

**Strengths:**

- The paper provides a comprehensive review of related work and connects to multiple research threads on the welfare implications of algorithmic decisions and competition between data-driven platforms.
- The authors apply their framework to three concrete binary classification setups that vary representation quality along different axes, demonstrating the usefulness of their approach.
- The paper is well-written and organized, making it easy to follow the authors' arguments and contributions.

**Weaknesses:**

- The paper would benefit from a more thorough evaluation of the proposed framework, including experiments on real-world datasets and comparison to existing methods.

- The paper could benefit from a more detailed discussion of the implications of their findings for real-world decision-making scenarios.


**Questions:**

- How does the quality of data representation affect the equilibrium social welfare in the authors' model, and how does this compare to existing models?

**Limitations:**

Yes.

---

> ### Author Rebuttal · Authors · 2023-08-10
>
> We thank the reviewer for their feedback. We present **new results** that address some concerns and also respond to other concerns.
>
> **“The paper would benefit from a more thorough evaluation of the proposed framework, including experiments on real-world datasets.**
>
> Beyond the experiments in the paper on binary classification on CIFAR-10 (Section 4.1) and synthetic data (Sections 3.2 and 4.2), we provide **new experiments with CIFAR-10 on 10 classes** (see Figure N1, General Response) and **new experiments for local improvements to data representations** (see Figure N2, General Response). We also have included some **more values of m in the original binary classification task** (see Figure N3 in pdf attached in the General Response). We observe non-monotonicity of the equilibrium social loss in these additional settings.
>
> **“The paper could benefit from a more detailed discussion of the implications of their findings for real-world decision-making scenarios.”**
>
> Our work suggests  that the consistent trend towards increases in scale might not lead to improvements in social welfare (the overall predictive accuracy for users) in today’s multi-platform settings. “Overall predictive accuracy”, which is the typical metric studied in the existing work on competing predictors (e.g. Ben-Porat and Tenneholtz [2017, 2019]), captures how well the overall marketplace of model-providers actually serves users, which is ultimately what is important from a societal perspective (e.g., Kleinberg and Raghavan 2021). The fact that scale may not reliably improve social welfare under competition should inform the emerging discourse on competition policy in digital marketplaces (e.g. Jullien and Sand-Zantman, 2020) and regulation/deployment of large models more broadly.
>
> Moreover, our results highlight the need for the research community to analyze scaling laws in competitive settings such as today’s multi-platform marketplaces.
>
> **“How does the quality of data representation affect the equilibrium social welfare in the authors' model, and how does this compare to existing models?”**
>
> Most of our results—Proposition 1, the results in Section 3.2, the results in Section 4.1, and the results in Section 4.2—aim to study the impact of data representation quality (as measured by Bayes risk) on the equilibrium social welfare. These results illustrate that improving data representation quality as measured by Bayes risk (e.g., from increases to scale) can decrease the *overall predictive accuracy* for users (a measure of social welfare).
>
> As to how our results compare to existing models, our paper is the first to study the impact of improving data representation quality (Bayes risk) on the equilibrium social welfare. Thus, there are no other clear models to compare our results with.
>
> Please let us know if this answers your question, we would be happy to clarify this further during the discussion period.

---

### Official Review · Reviewer_Gqjx · 2023-07-12

**Soundness:** 4 excellent
**Presentation:** 4 excellent
**Contribution:** 3 good
**Rating:** 7
**Confidence:** 2

**Summary:**

This paper looks and trends that arise in predictive utility across a population as model providers adapt to improving representations. At the core of the paper is an analysis that characterizes the population utility of equilibria as the representation varies in quality, noise, and dimension, which shows that overall utility may not be monotone along these axes. The paper additional contains an empirical study, over synthetic data and CIFAR, which addresses some of the simplifications made in the analysis while showing qualitatively similar trends in overall utility.

**Strengths:**

Topically, this paper addresses an interesting and potentially important question: what is the right way to model the effect that continued improvements to predictive accuracy have on social welfare? Some examples of prior work have pointed to a misalignment between predictive accuracy and social welfare, but the results in this paper are original and distinct, the the best of my knowledge. Namely, "model monoculture" can emerge when competing providers behave rationally, and representations are sufficiently good, leading to a subpopulation that isn't served by any provider.

Many of the concerns that I had about the simplified setting when going through the theoretical analysis were addressed to some extent in sections 3.3 and 4, and it seems like the results may be fairly robust to these choices.

The writing is polished, clear, and concise.

**Weaknesses:**

A central assumption of the work is that all model providers have access to the same global representation. This makes some sense, but is not all that realistic: despite using on the same underlying techniques and relying on the same public-knowledge advances in learning, different model providers show very different performance metrics. The market reputation factor incorporated in 3.3 does not quite address this, as it is independent of model performance.

While the experimental analysis is a nice demonstration that the theoretical results hold in some more realistic settings, there are still some simplifications that weren't explicitly motivated (only linear classifiers, binarized CIFAR). These choices make sense in that they are consistent with the rest of the paper, but the point of the experimental analysis is to show that the results are robust, so they do limit the significance of the work.

**Questions:**

1. If you are able, please explain the rationale behind the global representation used in your work. To what extent do you believe that your results depend on this assumption?
2. Do you have any insight into the likely outcomes if your experiments were to relax some of the conditions that I mentioned above?

**Limitations:**

The section 5 at the end of the paper would be a good place to add discussion of the fact that there is certain risk that the results are overly-sensitive to choices made in modeling and experiment design. It currently does not, and limitations are not discussed in much detail elsewhere.

---

> ### Author Rebuttal · Authors · 2023-08-10
>
> We thank the reviewer for their feedback. We present **new results** that address some concerns and also respond to other concerns.
>
> **“There are still some simplifications that weren't explicitly motivated (binarized CIFAR).”**
>
> We provide new results showing that our empirical results in Section 4.3 directly extend to CIFAR-10 with 10 classes, and our theoretical results in Section 3.1 directly extend to multi-class classification with K classes. (Our motivation for studying binary classification was to simplify the exposition and analysis.)
>
> In **new multi-class experiments on CIFAR-10**, we consider a data distribution over 50,000 samples and the original 10 classes. We show that the equilibrium social loss is non-monotonic in the data representation quality as measured by Bayes risk (see Figure N1 in the General Response pdf). Figure N1 shows that the empirical results in Section 4.3 directly extend to CIFAR-10 with 10 classes.
>
> In a new theoretical result, we show that Proposition 1 can be directly extended to multi-class classification, where the function class is all functions $f:X \rightarrow Y$ where $Y = [K]$. The same insights from Section 3.1 thus carry over to multi-class classification.
>
> For details of the new experiments and new theoretical result, we refer the reviewer to the "Extension to classification with more than 2 classes" section of the General Response.
>
> We also extend our theoretical result (Proposition 1) to classification with K classes. We refer the reviewer to the General Response for details.
>
> **“To what extent do you believe that your results depend on this assumption [the global representations assumption]?”**
>
> We provide **an additional CIFAR-10 experiment to study “local improvements”** each model-provider uses different data representations and where only one model-provider changes their data representations. We continue to observe non-monotonicity of the social loss (see Figure N2 in the attached pdf). In these setups, data diversity also impacts the equilibrium social loss: that is, changing a single model-provider’s data representations can increase (or decrease) the overall diversity in the data representations across model-providers, which we expect would decrease (or increase) the level of monoculture/outcome homogeneity. Understanding exactly how varying data representations impacts data diversity (and equilibrium social loss) is an interesting direction for future work.
>
> **“If you are able, please explain the rationale behind the global representation used in your work.”**
>
> As described above, we provide an additional experiment on CIFAR-10 to study “local improvements” where model-provider use different data representations and where only one model-provider changes their data representations. We continue to observe non-monotonicity of the social loss (see Figure N2 in the attached pdf).
>
> The reason that we focused on global data representations is to isolate the role of competition on the social loss. In particular, we intentionally designed our model to be as close as possible to the classical single-decision-maker setting (where there is a single data distribution), so that we could directly compare scaling trends in the single decision-maker setup and the competitive setup. Our results show that competition leads to non-monotonicity in the social loss, even if the model-providers are given the exact same data representations and have access to the same function class.
>
> One real-world motivation for global data representations is that the model-providers all rely on the same pretrained model for representations, but each model-provider fine-tunes their model separately.
>
> **“There are still some simplifications that weren't explicitly motivated (only linear classifiers).”**
>
> Let’s return to the real-world motivation described above, where all model-providers all rely on the same pretrained model for representations, but each model-provider fine-tunes their model separately. We view the restriction to linear functions as a stylized way of capturing finetuning.
> More broadly, considering nonlinear predictors is interesting, but makes it more challenging to compute the Nash equilibria via gradient-descent-based best response dynamics.
>
>
> **“The section 5 at the end of the paper would be a good place to add discussion of the fact that there is certain risk that the results are overly-sensitive to choices made in modeling and experiment design.”**
>
> Using the extra space in the final version of the paper, we will add a qualitative discussion of our assumptions that highlights the simplifications in the user choice formalization, market reputations formalization, and model-provider actions formalization. We hope that our results inspire future work that extends our model to relax these assumptions and incorporate additional complexities.

---

> > ### Comment · Reviewer_Gqjx · 2023-08-16
> > **Thank you for your rebuttal**
> >
> > Thank you for putting together a very detailed rebuttal! The additional data on CIFAR10, and the generalized version of Proposition 1, do address some of my concerns about the setting, and your experiments with local representations seem to illustrate the need for additional work. I look forward to seeing the final version of this paper, and will maintain my score and support for it.

---

### Official Review · Reviewer_5U8U · 2023-07-26

**Soundness:** 2 fair
**Presentation:** 3 good
**Contribution:** 3 good
**Rating:** 6
**Confidence:** 3

**Summary:**

This paper shows that "social welfare" might (under some settings) be worse-off as the representation quality of machine learning models increase when there are multiple strategic machine-learning-model providers.

The paper focuses the analysis on binary classification tasks. The paper assumes that users choose the model provider which gives the most accurate prediction. And multiple strategic machine-learning-model providers compete for the number of users choosing them. The social welfare is defined as the difference between the most accurate model prediction and the true label in expectation. The representation quality is defined as the Bayes risk. The paper shows that under a stylized setting, increasing the representation quality might decrease the social welfare.

Then many empirical evaluations are conducted to show that this also happens in several relaxed settings.






**Strengths:**

The paper is well-written and easy to follow.

The paper tries to answer an interesting question of what is the effect of representation quality on social welfare when there are competing machine-learning-model providers.

The paper provides theoretical analysis on a stylized setting which shows an intriguing phenomenon: when the representation quality increases, the defined social welfare can decrease. And it also provides empirical evaluations to show this phenomenon beyond the restricted setting. This calls for further discussion on whether this happens more broadly and how to address this problem.

**Weaknesses:**

My major concern about this paper is that all the theoretical analysis and empirical evaluation are conducted on settings where all model providers have the same representation. In reality, competing providers do not share their models and they rarely have the same representations. So the conclusion of this paper is based on evidence on a very restricted setting.

I am not sure whether binary classification is the right problem to analyze the concerned question. First, if there are only two options, the users can simply look at the two options and make a decision. They do not even need to hop between model providers. Second, in the examples given in the paper, there is no single right choice of prediction (e.g. there might be multiple relevant webpages and songs, routes, and chat messages).

Minor:

In the description of Figure 1, "in the left plot than the plot plot" -> "in the left plot than that in the right plot".







**Questions:**

See weaknesses.

**Limitations:**

To me, the authors adequately addressed the limitations.

---

> ### Author Rebuttal · Authors · 2023-08-10
>
> We thank the reviewer for their feedback. We present **new results** that address some of their concerns and also respond to other concerns below.
>
> **“I am not sure whether binary classification is the right problem to analyze the concerned question.”**
>
> We provide new results showing that our empirical results in Section 4.3 directly extend to CIFAR-10 with 10 classes, and our theoretical results in Section 3.1 directly extend to multi-class classification with K classes. (Our motivation for studying binary classification was to simplify the exposition and analysis.)
> - In new experiments on CIFAR-10, we consider a data distribution over 50,000 samples and the original 10 classes. We show that the equilibrium social loss is non-monotonic in the data representation quality as measured by Bayes risk (see Figure N1 in the General Response pdf). Figure N1 shows that the empirical results in Section 4.3 directly extend to CIFAR-10 with 10 classes.
> - In a new theoretical result, we show that Proposition 1 can be directly extended to multi-class classification, where the function class is all functions $f:X \rightarrow Y$ where $Y = [K]$. The same insights from Section 3.1 thus carry over to multi-class classification.
>
> For details of the new experiments and new theoretical result, we refer the reviewer to the "Extension to classification with more than 2 classes" section of the General Response.
>
> **"My major concern about this paper is that all the theoretical analysis and empirical evaluation are conducted on settings where all model providers have the same representation."**
>
> We provide an **additional experiment to study “local improvements”** where each model-provider uses different data representations and only one model-provider changes their data representations. We continue to observe non-monotonicity of the social loss (see Figure N2 in the attached pdf) in a CIFAR-10 setup. In these setups, data diversity also impacts the equilibrium social loss: that is, changing a single model-provider’s data representations can increase (or decrease) the overall diversity in the data representations across model-providers, which we expect would decrease (or increase) the level of monoculture/outcome homogeneity. Understanding exactly how varying data representations impacts data diversity (and equilibrium social loss) is an interesting direction for future work.
>
> **“In reality, competing providers do not share their models and they rarely have the same representations.”**
>
> As described above, we provide an additional experiment on CIFAR-10 to study “local improvements” where only one model-provider changes their representations. We continue to observe non-monotonicity of the social loss (see Figure N2 in the attached pdf).
>
> The reason that we focused on global data representations is to isolate the role of competition on the social loss. In particular, we intentionally designed our model to be as close as possible to the classical single-decision-maker setting (where there is a single data distribution), so that we could directly compare scaling trends in the single decision-maker setup and the competitive setup. Our results show that competition leads to non-monotonicity in the social loss, even if the model-providers are given the exact same data representations and have access to the same function class.
>
> One real-world motivation for global data representations is that the model-providers all rely on the same pretrained model for representations, but each model-provider fine-tunes their model separately. This also motivates our consideration of model-providers choosing linear functions over the shared representations in our experiments.

---

> > ### Comment · Reviewer_5U8U · 2023-08-16
> >
> > The additional theoretical analysis and exp addressed some of my concerns. I would like to keep my evaluation since I still think that the setting (for the theoretical analysis) where all models share the same representation is slightly limited.

---

### Official Review · Reviewer_4Cpq · 2023-07-31

**Soundness:** 3 good
**Presentation:** 3 good
**Contribution:** 3 good
**Rating:** 5
**Confidence:** 3

**Summary:**

The authors consider a setting where there exist multiple provides of a machine learning model, and all together choose to increase their representation quality. The authors show, via both theory and experiment, that this improvement in data representation does not necessarily translate to improved social welfare. In particular, the authors shows that, due to their model of competition,  when data representation qualities increases, there is less specialization in that the models are unlikely to differ significantly in providing strong performance for a certain group of users, and the equilibrium achieved may be worse off in terms of overall loss each user received. The authors present empirical results on CIFAR-10 using varying pretrained representations, as well as a generalization of their theory to situations where users break tied based on model-provider reputation.

**Strengths:**

Overall, I think the paper studies a very interesting setting (multiple machine learning providers in competition), that is increasingly relevant these days (though I think any model of this should at least strive to capture the more realistic setting in that some players, like Google/OpenAI/Apple are clearly very dominant in their respective spaces, and at times might be seen more as monopolies,  and not just from a market-reputation `tie-breaking' angle).

The paper is also well-written and easy to follow, and has a good mix of theoretical and empirical results.



**Weaknesses:**

My main concerns with the paper are lack of discussion on the assumptions needed, and confusion about the central takeaway of the work re: its relationship to the scaling law literature.

W1 Scaling Laws / Takeaway: Even after a few reads, the framing of the work as a contrast to current works on scaling laws that only focus on one model feels a bit arbitrary, and it seems a bit unfair to say the focus on one-model is a "limitation" of work on scaling laws. I don't agree that the purpose of scaling law research is to make a statement on social welfare, and rather to precisely characterize phenomena that change as you increase model size / parameters. I think focusing on clean measures like loss as # parameters / depth / etc. are important to rigorously do, as for a lot of research in the field its purpose is to shed some empirical insights on the model and training processes themselves.  Whereas this work opens up so many questions about the right way to model social welfare, user choice, model provider decisions, etc., I feel like it is a big departure from the intention of most research studying scaling trends.

I think at minimum, the authors should try to soften claims where they draw this contrast, such as the first two sentences of the abstract, that almost seem to imply that the finding that accuracy improves with scale is only due to the one-model, when what the authors measure (social welfare) is anyways different from accuracy. I am mainly concerned that this paper runs the risk of misleading general readers who might just get the takeaway "we shouldn't try to increase model size". That might be an attention-grabbing claim, but I don't think the small-scale setting the authors study warrants such a claim - at best, it opens up the need for more research on analyzing competition in the "large-scale machine learning models" space.

Therefore, I think the authors should also think carefully what the central takeaway of the paper really is. Even if the authors stick with the contrast with scaling trends as a broad framing, I'm still not sure its too strong/surprising of a finding.

W2: Why is keeping the data constant a realistic assumption to make? None of the cited examples to motivate the paper (chatGPT vs. Claude, Spotify v. Pandora) have this property, particularly when you consider that their data is reliant on the users who use their serve, as well as methods like instruction-tuning and RLHF that are more proprietary and key to the mass deployment of these models. in the LM space, we can see this with the development of products like CharacterAI. In a competitive setting, if model providers could change the data they use, wouldn't that be an action they could take to provide unique services that would break the setting studied by the authors?
It seems like the purpose of the assumption is only to be able to focus on the scaling trends, which goes back to W1. But at the least, this assumption should be discussed.

W3: I think the reasons behind why increasing data quality representations leads to models that are "likely to agree with each other on most data points" should be clearly stated. Isn't one aspect important for social welfare whether it is a particular kind of user subgroup that is hurt by all models (related to the cited work on homogenization)? The data representation methods studied (e.g. varying the representation dimension, representation noise)  are all pretty synthetic that don't result in semantically meaningful improvements (e.g. over a subgroup) in a way more realistic improvements (increase deep NNs model parameters) might.

W4. While I think the generalization to asymmetry in the market where model-provider reputation matters is a great addition, this still feels somewhat limited (only executed through the form of weighting when breaking ties). People choose models based on a variety factors including ease of service, its performance over the generally distribution of inputs the user expects to query with, etc. -- why pick just reputation in a limited way to study? Again, I think these limitations should at least be stated.

Minor

- I think more experiment details should be included in the main paper, such as calculating the Bayes risk using gradient descent from pretrained representations for the CIFAR10 experiments. Since the paper's message seems intended to capture a setting that is only really arising due to large, pre-trained models that afford high-quality data representations, I think these experimental results are important to be clear to the community about, and I might suggest even including another experiment in the language domain.

- The abstract could hint at the punchline "why" for the decrease in social welfare (the fact with low quality representations there is a more  heteregenous set of choices)

- typo in line 162

**Questions:**


- What are the bars on Figure 3? Standard error or CIs?

- Did the authors try more values of m? Is there a trend to observe over m? Currently, it is hard to make an overall statement just comparing m=8 and 12.

- Line 263: Should this say initialize randomly? How important is this choice?

**Limitations:**

There is essentially no limitations section in the paper, which I feel is important given the many assumptions used for overall toy setting that aren't reflected in the real world the authors motivate the work with. While the authors did address prior work limitations by generalizing to a setting where users are influenced by model provider reputation, even this is only accounted for in the tie-breaking scheme and I think the overall user choice setting is quite idealized.
Furthermore, in practice users would likely have several different inputs, and pick the model that provides low loss on the first few inputs they try with - this suggests consider the distribution of inputs different users care about, which the authors' model doesn't study.

I think the authors should state their assumptions very clearly, and explain why running additional experiments / analyzing settings without these assumptions are out of scope for this work.

---

> ### Author Rebuttal · Authors · 2023-08-10
>
> We thank the reviewer for their feedback. We present **new results** that address some concerns and also respond to other concerns.
>
> **“Additional experiments / analyzing settings without these assumptions.”**
>
> We provide new CIFAR-10 experiments on 10 classes (Figure N1, General Response) and for local improvements to data representations (Figure N2, General Response). We consistently observe non-monotonicity, which provides evidence that our conclusions are robust to different assumptions/settings.
>
> **“Did the authors try more values of m? Is there a trend to observe over m?”**
>
> We computed additional values of $m$ for the original binary classification task (see Figure N3 in General Response pdf). Non-monotonicity is consistently observed. Interestingly, the curve shape changes with $m$, which aligns with our theoretical findings (Proposition 1).
>
> **“What the central takeaway of the paper really is.”**
>
> The primary takeaway (lines 35-36) is that improvements to data representation quality can decrease the *overall predictive accuracy* for users (i.e., social welfare). A consequence for scaling laws is that scaling trends for predictive accuracy of a single model-provider may not capture scaling behavior for *overall predictive accuracy* for competing model-providers, which is important when considering the impact of large-scale ML systems on society.
>
> **“I feel like it is a big departure from the intention of most research studying scaling trends.”**
>
> Our work aims to bring new perspectives from the emerging discourse on platform competition and policy in digital marketplaces (e.g. Jullien and Sand-Zantman, 2020) into the analysis of scaling trends. The prevalence of competition in today’s multi-platform settings motivates studying scaling trends under competing model-providers, which indeed goes beyond the scope of most research on scaling laws.
>
> As to why we focus on social welfare (which differs from typical metrics in the scaling laws literature), we believe that social welfare (i.e. “overall predictive accuracy”) is a more natural metric for the multiple model-providers setting. “Overall predictive accuracy” is studied in prior work on competing predictors (e.g. Ben-Porat and Tenneholtz [2017, 2019]) and captures how well the marketplace serves users, which matters for society (e.g., Kleinberg and Raghavan 2021).
>
> Our results provide some evidence that the impact of increased scale can significantly differ in multi-platform and in single-platform settings, making the study of how competition impacts scaling trends an important avenue of future research.
>
> **“I don't agree that the purpose of scaling law research is to make a statement on social welfare. It seems a bit unfair to say the focus on one-model is a "limitation" of work on scaling laws.”**
>
> We agree with the reviewer that the focus on a single model is not a “limitation” of scaling laws and we do not intend to claim that anywhere in the paper. We will make this clear in the final version and clarify that we study a different setting (competing model-providers) and a different metric (social welfare) than the typical scaling laws literature. If the reviewer has a specific sentence in mind, please let us know which one.
>
> **“This work opens up so many questions about the right way to model social welfare, user choice, model provider decisions. Many assumptions used for overall toy setting that aren't reflected in the real world.”**
>
> We agree that the multi-platform setting is inherently messier to model than the single platform setting. Our work is the first to study how competition between model-providers impacts social welfare, and introducing simplifying assumptions was necessary to make progress on analyzing the equilibria. Despite these assumptions, we expect the primary takeaway of our work (that social welfare can be non-monotonic in improvements to data representations) to be robust.
>
> **“There is essentially no limitations section in the paper.”**
>
> We will include a more thorough discussion of our assumptions in the final version. We hope that our results inspire future work that incorporates additional complexities (e.g., richer user choice models) and that carries out additional empirical evaluation (e.g., for language models).
>
> **“I think the reasons behind why increasing data quality representations leads to models that are "likely to agree with each other on most data points" should be clearly stated.”**
>
>  For high-quality representations, a Bayes optimal predictor is incorrect only on a small set of users (call this set $S$). Intuitively, any predictor that correctly labels users in $S$  would mislabel many other users (e.g., if the Bayes optimal label for $x$ is $1$, correctly labeling users with $(x, 0)$ requires mislabelling the many other users with $(x, 1)$). This could happen in practice when representations fail to encode a predictive feature. If representation quality is high, a model-provider optimizing for market share would choose the Bayes optimal label, regardless of other models’ providers actions (e.g., if $p(y = 1|x) = 0.9$ and $m=3$, it’s always better to choose $f(x)=1$ and get $\ge 0.33$ market share than choose $f(x) = 0$ and get $\le 0.1$ market share). This suggests model-providers will "agree" on many data points at equilibrium. With lower quality representations (e.g., $p(y = 1|x) = 0.6$), the tradeoff would change, which could lead to “disagreement”. Proposition 1 mathematically formalizes this intuition.
>
> **“I think more experiment details should be included in the main paper.”**
>
> We will add more experiment details to the final version, by expanding upon and moving some of Appendix B into the main body.
>
> **“Line 263: Should this say initialize randomly?”**
>
> We initialized to a $0$-mean gaussian with stdev 0.1. The stdev needs to be sufficiently high for convergence of best-response dynamics.
>
> **Bars on Figure 3:**
>
> Bars are based on standard errors.

---

> > ### Comment · Reviewer_4Cpq · 2023-08-18
> > **Response**
> >
> > Thank you for your response -- most of my questions were answered.
> >
> > Regarding specific sentences that relate this work to the scaling law literature -- I believe even the abstract and introduction frame this work as addressing a limitation of scaling laws (i.e. "However, these scaling laws typically take the perspective..." , "in reality providers often compete.."). One could instead frame the introduction entirely about studying multi-platform settings, and then motivate why it would be useful to study the effect of scale in this setting. I still believe the current framing is a bit misleading -- I think this paper holds more value to a researcher focusing on multi-platform markets than scaling laws (as this paper wouldn't fundamentally change the core research question people studying the latter focus on), so in the revised version I encourage the authors to increase clarity of their paper in this regard.
> >
> > It would be great if the authors provided in their rebuttal the limitations they intend to expand on, instead of just stating they will include a section!

---

> > > ### Author Response · Authors · 2023-08-19
> > >
> > > Thanks to the reviewer for reading our response and asking follow-up questions. We respond below and are happy to elaborate further on any of these points if the reviewer has additional questions.
> > >
> > > **Limitations:** Here is a list of limitations we intend to expand on:.
> > > - *Simplicity of user choice model*: Building on previous work on competing platforms (e.g., Ben-Porat and Tennenholtz [2017, 2019]), our user choice model assumes a simplified and stylized discrete choice model where each user individually (and noisily) chooses the model-providers that offers them the best predictive accuracy on a *single input*. Many of our results thus assume that a user’s choice of platform is fully specified by the platforms’ choices of predictor (i.e. platforms are ex-ante homogeneous). Some of our results (Section 3.2) relax this assumption by introducing “market reputations” (modeled as global weights in the logit model for discrete choice); however, this formalization assumes that market reputations are global across users and that market reputations surface as tie-breaking weight in the noiseless limit.
> > > - *Assumptions on model-provider action space*: Building on Ben-Porat and Tennenholtz [2017, 2019], we also assume that the only action taken by model-providers is to choose a classifier from a prespecified class. This formalization does not capture other actions (such as data collection and price setting) that may be taken by the platform.
> > > - *Specificity of the task*: Our work focuses on multi-class (see rebuttal) classification tasks with a discrete, unstructured output space. This formalization does not capture generative tasks with image-based or text-based outputs.
> > > - *Global representations assumption*: Our results assume that all model-providers share the same representation and improvements to representations are experienced by all model-providers. This setup of global data representations is motivated by a marketplace where all model-providers all rely on the same pretrained model for representations, but each model-provider separately fine-tunes their own model. However, this framework does not allow for heterogeneity or local improvements in the data representations. (That being said, we provide experiments (see rebuttal) for a setup where each model-provider uses a different data representation and only one model-provider updates their representations, which provides preliminary evidence that non-monotonicity continues to be exhibited in greater generality.)
> > >
> > > Our work is the first to study how competition between model-providers impacts social welfare, and introducing simplifying assumptions was necessary to make progress on analyzing the equilibria. Despite these assumptions, we expect the primary takeaway of our work (that social welfare can be non-monotonic in improvements to data representations) to be robust.
> > >
> > > Note that our experiments from the rebuttal for *multi-class classification* and *local improvements to heterogeneous representations* already illustrate the robustness of results to some of these assumptions.  We hope that future work relaxes other  assumptions to further bridge the gap between our stylized model and real-world digital marketplaces.
> > >
> > > **Value of our results for the scaling laws community:** It is true that the scaling law community has yet to deeply engage with broader societal implications of increasing the scale of models in the presence of multiple model providers. This is why we disagree with the reviewer’s assessment that our work is of  limited value to the scaling laws community. Given that our results pinpoints a distinction between scaling trends in the single-platform setting and multi-platform competitive settings, we hope our results encourage the scaling laws community to expand the scope of their current research questions to examine the impacts of scale in multi-platform, competitive settings, even though the multi-platform setting is not currently the focus of scaling laws research.
> > >
> > > More broadly, we hope that our work encourages both the scaling laws community and the platform competition community to synthesize their perspectives and contribute to the emerging, interdisciplinary area on the societal impacts of competing ML systems.

---

### Author Rebuttal · Authors · 2023-08-10

Thanks to the reviewers for their detailed feedback. In response to reviewer feedback, we provide the following new results: (1) we provide **additional experiments on CIFAR-10 and theoretical results for multi-class classification** and (2) we provide **additional experiments for non-global representations** across model-providers. (We respond to other points raised by the reviewers in the individual responses to each reviewer.)

## Extension to classification with more than 2 classes
Several reviewers asked if our results extend beyond binary classification. We provide new experiments showing that our empirical results in Section 4.3 directly extend to CIFAR-10 with 10 classes, and new theoretical results showing our results in Section 3.1 directly extend to multi-class classification with $K$ classes.

**Extension of empirical results on CIFAR-10:** In *new CIFAR-10 experiments for multi-class classification*, we consider a data distribution over 50,000 samples and the original 10 classes. Each function maps each x to a probability distribution $p$ over the 10 classes, according to a linear function followed by a softmax. The loss function $\ell(y, p)$ is equal to $|1 - p(y)|$, where $p(y)$ is the probability of choosing the correct class $y$. We show that the equilibrium social loss is non-monotonic in the data representation quality as measured by Bayes risk (see Figure N1 in the General Response pdf). Figure N1 shows that the empirical results in Section 4.3 directly extend to CIFAR-10 with 10 classes.

**Extension of theoretical results (Proposition 1):** In a *new theoretical result for multi-class classification*, we show that Proposition 1 can be directly extended to $K$-class classification. The function class is all functions $f:X \rightarrow Y$ where $Y = [K]$.

*Result*: Let $X$ be a finite set. For each class $i \in [K]$, let $\alpha_i(x) := \mathbb{P}[Y = i | X = x]$ be the probability of each class for the representation $x$. At any pure strategy equilibrium, the equilibrium social loss is in between $E[\sum_{i \in [K]} \alpha_i(x) \cdot  I[\alpha_i(x) < 1/(2m)]]$ and $E[\sum_{i \in [K]} \alpha_i(x) \cdot I[\alpha_i(x) \le 1/m]]$.

*Comparison with Proposition 1*: Note that $\alpha(x)$ in the paper is equal to $\min(\alpha_0(x), \alpha_1(x))$. In the special case of 2 classes, this result exactly corresponds to Theorem 1 with a factor of 2 slack in the indicator random variable.

*Takeaway*: When representations improve—and thus the $\alpha_i(x)$ values approach either $0$ or $1$— we see that the equilibrium social loss can increase. The same insights from Section 3.1 thus carry over to multi-class classification.

*Proof sketch*: The proof of this result follows a very similar argument to the proof of Proposition 1. It suffices to show (a) if $\alpha_i(x) > 1/m$, then some model-provider will choose $f(x) = i$, and (b) if $\alpha_i(x) < 1/(2m)$, then no model-provider will choose $f(x) = i$. To see this, like in the proof of Proposition 1, we can restrict to the case where $X$ only contains a single data representation $x$ (apply a generalization of Lemma 3 in Appendix C). The first part follows from the fact that some model-provider must get market share $\le 1/m$ so that model-provider would want to switch to label $i$ if no other model-provider chooses $i$. The second part follows from the fact that some model-provider gets market share $\ge 1/m$, and any model-provider can get market share $\ge 1/(2m)$ by copying the label of that model-provider.

## Beyond global representations across model-providers

**Extension of empirical results on CIFAR-10:** To address the reviewers’ questions about differing representations, we also provide *a new experiment on CIFAR-10 to study “local improvements”* where each model-provider uses different data representations and only one model-provider changes their data representations. We continue to observe non-monotonicity of the social loss (see Figure N2 in the attached pdf) in a CIFAR-10 setup. In these setups, data diversity also impacts the equilibrium social loss: that is, changing a single model-provider’s data representations can increase (or decrease) the overall diversity in the data representations across model-providers, which we expect would respectively decrease (or increase) the level of monoculture/outcome homogeneity. Understanding exactly how varying data representations impacts data diversity (and equilibrium social loss) is an interesting direction for future work.

**Discussion of global representations assumption:** The reason that we focused on global data representations is to isolate the role of competition on the equilibrium social loss. In particular, we intentionally designed our model to be as close as possible to the classical single-decision-maker setting (where there is a single data distribution), so that we could directly compare scaling trends in the single decision-maker setup and the competitive setup. Our results show that competition leads to non-monotonicity in the social loss, even if the model-providers are given the exact same data representations and have access to the same function class.


In addition, in terms of real-world motivation, global data representations capture a marketplace where all model-providers all rely on the same pretrained model for representations, but each model-provider separately fine-tunes their own model. In fact, the linear function class in our experiments can be viewed as a stylized model for fine-tuning.

---

### Decision · Program_Chairs · 2023-09-21

**Decision:**

Accept (poster)

**Comment:**

This paper studies an interesting problem in which improved representations (in terms of Bayes risk) result in reduced social welfare under competition. All reviewers found the paper to be clear, organized, and well-written, and appreciated the paper's complementing mix of theory and empirics. The general message of this paper is interesting and relevant, showing how competition can challenge some of our fundamental beliefs on scaling processes. The authors' efforts in adding results (multi class and "local" representations) in response to reviewer requests is commendable.

Reviewers were overall positive, but nonetheless pointed our several issues that we hope the authors would take into account in their final version:
1. Reviewers felt that the limitations of the approach and the implications of its assumptions could have been better discussed. For this the authors provided list of additions, which we hope they will implement.
2. Several reviewers found the assumption of global representation to be insufficiently justified. The additional experiment on local representations helped, but this should not be considered a substitute for a proper justification of the main assumption.
3. Some reviewers felt that the market setting should be better justified, and have asked several "what if" questions that challenge the paper's main results. Authors are encourage to address these.
4. One reviewer found the paper's framing around "scaling laws" to be incompatible with how the actual scaling laws community views these. Authors are encouraged to make the necessary adaptations so that their message will be relevant to a broader audience, including this community.

One final point regards the paper's title - which is perhaps somewhat misleading, since it implies that what causes a reduction in welfare is the improvement of Bayes risk, where the actual control variable is representation quality (there are other ways to improve risk). If the authors agree with this, we hope they would be willing to reconsider the message they would like to convey through their title.